# Understanding Land–Atmosphere–Climate Coupling from the Canadian Prairie Dataset

**Alan K. Betts** [1,]*  **and Raymond L. Desjardins** [2]

[1] Atmospheric Research, Pittsford, VT 05763, USA
[2] Agriculture and Agri-Food Canada, Ottawa, ON K1A0C6, Canada; ray.desjardins@canada.ca
[*] Correspondence: akbetts@aol.com; Tel.: +1-802-483-2087

**Abstract:** Analysis of the hourly Canadian Prairie data for the past 60 years has transformed our quantitative understanding of land–atmosphere–cloud coupling. The key reason is that trained observers made hourly estimates of the opaque cloud fraction that obscures the sun, moon, or stars, following the same protocol for 60 years at all stations. These 24 daily estimates of opaque cloud data are of sufficient quality such that they can be calibrated against Baseline Surface Radiation Network data to yield the climatology of the daily short-wave, long-wave, and total cloud forcing (SWCF, LWCF and CF, respectively). This key radiative forcing has not been available previously for climate datasets. Net cloud radiative forcing changes sign from negative in the warm season, to positive in the cold season, when reflective snow reduces the negative SWCF below the positive LWCF. This in turn leads to a large climate discontinuity with snow cover, with a systematic cooling of 10 °C or more with snow cover. In addition, snow cover transforms the coupling between cloud cover and the diurnal range of temperature. In the warm season, maximum temperature increases with decreasing cloud, while minimum temperature barely changes; while in the cold season with snow cover, maximum temperature decreases with decreasing cloud, and minimum temperature decreases even more. In the warm season, the diurnal ranges of temperature, relative humidity, equivalent potential temperature, and the pressure height of the lifting condensation level are all tightly coupled to the opaque cloud cover. Given over 600 station-years of hourly data, we are able to extract, perhaps for the first time, the coupling between the cloud forcing and the warm season imbalance of the diurnal cycle, which changes monotonically from a warming and drying under clear skies to a cooling and moistening under cloudy skies with precipitation. Because we have the daily cloud radiative forcing, which is large, we are able to show that the memory of water storage anomalies, from precipitation and the snowpack, goes back many months. The spring climatology shows the memory of snowfall back through the entire winter, and the memory in summer, goes back to the months of snowmelt. Lagged precipitation anomalies modify the thermodynamic coupling of the diurnal cycle to the cloud forcing, and shift the diurnal cycle of the mixing ratio, which has a double peak. The seasonal extraction of the surface total water storage is a large damping of the interannual variability of precipitation anomalies in the growing season. The large land-use change from summer fallow to intensive cropping, which peaked in the early 1990s, has led to a coupled climate response that has cooled and moistened the growing season, lowering cloud-base, increasing equivalent potential temperature, and increasing precipitation. We show a simplified energy balance of the Prairies during the growing season, and its dependence on reflective cloud.

**Keywords:** climate; land–atmosphere interaction; clouds; diurnal cycle; snow cover; Prairies; land-use; hydrometeorology

## 1. Introduction

Understanding land–atmosphere–climate coupling is challenging, because so many coupled processes are involved: soil temperature and moisture, vegetation types, properties and coverage, near-surface temperature and humidity, the atmospheric boundary layer, the shallow and deep cloud fields which determine the surface radiation balance and surface precipitation, and the soil hydraulic properties that determine the surface and deep runoff, to name only the local components. In the cold season, precipitation falls as snow, and the surface accumulation increases the albedo, and stores water until snowpack melt.

The coupling between the energy and water cycles at the land surface is central to hydrometeorology, and important to weather forecasts on timescales from days to seasons. Earlier reviews [1,2] looked at hydrometeorology from the global modeling perspective using model reanalysis data, which showed how net long-wave and short-wave radiation, cloud cover, surface fluxes, diurnal temperature range, soil moisture, and cloud-base height were coupled on daily timescales over river basins [3]. On daily timescales, the land–atmosphere system is fully coupled, so that errors in the model representation of processes in the soil, vegetation, boundary layer, and cloud fields can rapidly bias a model forecast. Nonetheless, this model perspective was a strong motivation for our analyses of the Canadian Prairie data, and the search for a quantitative description of the fully coupled observed system.

Historically, many climate and hydrometeorology studies have been largely based on precipitation, temperature, and humidity, for which long-term records are available [4–6]. However, the diurnal cycle is driven primarily by the surface radiation balance, which depends critically on the daily cloud fields, which are generally unknown in climate records, until satellite-based estimates became available. We cannot study the fully coupled nature of the land–atmosphere–climate system without the surface radiation budget.

The Canadian Prairie data is, however, an exception, because observers, typically at most major airports, were trained to estimate hourly the opaque cloud fraction in tenths, by cloud level and in total. The definition of opaque cloud is "opaque to the sun, moon, or stars"; and this protocol has been followed by trained observers hourly for 60 years across the Prairies. With 24 observations per day (almost none are missing), we have representative estimates of the fraction of the daytime short-wave clear-sky (SWCS) flux reaching the surface, and the fraction of the sky that is opaque to outgoing long-wave (LW) radiation for over 600 station-years of data. Because there are 17 years of Baseline Surface Radiation Network (BSRN) data just 25 km south of Regina, SK, we were able to calibrate the opaque cloud data in terms of the LW and SW cloud forcing (Section 3.3). This is transformative, as it meant that we were able to determine quantitatively the climate coupling between the cloud radiative forcing, and the diurnal and seasonal cycle. In addition, simply because we can separate the large radiative impact of clouds from the impact of precipitation, we can better quantify the hydrometeorological processes that couple the energy and water cycle, and observe the long-term memory of precipitation anomalies. In recent years, data from the Gravity Recovery and Climate Experiment (GRACE) [7,8] give estimates of the seasonal drawdown of total water storage. Canadian archives also record agricultural crops grown on the Prairies back to 1955, so we could assess the large climate impact of the shift away from summer fallowing to continuous cropping.

This paper is not a conventional review of the literature on land–atmosphere–climate coupling. Instead, it is a synthesis of our key conclusions from a series of Prairie data analysis papers [9–16]. Readers interested in more details, or in the evolution of our thinking, can refer back to these original papers. It is remarkable that the long-term Prairie climate dataset, with better cloud observations, have taken our understanding of land–atmosphere–climate coupling to a new level. In retrospect, much of our analysis could have been done two decades ago, but the data was not widely accessible.

Section 2 discusses the Prairie data and our analysis methods. Section 3 outlines how the climate is coupled to opaque cloud and snow cover on daily timescales, and shows the difference in cloud forcing between warm and cold season with snow. Section 4 looks at the long-term memory of

precipitation anomalies, both using multiple regression for the cold and warm season memory, and the dependence of the diurnal coupling on opaque cloud and precipitation anomalies. Section 5 looks at how the seasonal extraction of the surface total water storage dampens the interannual variability of precipitation anomalies in the growing season, and how the large land-use change from summer fallowing to intensive cropping has led to a coupled climate response. Finally, we return to reanalysis data to show how the growing season surface and top-of-atmosphere (TOA) budgets change with cloud cover. Section 6 summarizes our conclusions.

## 2. Methods

### 2.1. Prairie Station Locations

Figure 1 shows the location of the 15 Prairie stations used in our analyses. Most of the stations are in the agricultural region, except The Pas. Table 1 lists the station locations and elevation, and the two letter code is used to identify stations in the figures and text. These have an hourly pressure, temperature, relative humidity, wind speed and direction, opaque cloud, and derived radiation, starting in 1953, for all stations, except RG, and MJ, which start in 1954, and ED, which starts in 1961. We accessed the data through June 2011. The hours of missing data are remarkably small. For key stations, such as Calgary, Regina and Winnipeg, more than 99.9% of the days have no missing hours in the first 40 years. In more recent years since 1994, the number of days with less than 23 h of data is typically less than 1%. A few stations (PS in 1992; MJ in 1998; LE and MH in 2006) shifted to daytime-only observation in recent years, because of reduced staffing. The stations also have daily precipitation and snow depth (except for PO), although the last year with complete precipitation data was 1994 for SW, 2005 for LE and MH, 2007 for WI, 2008 for RG, and 2009 for SK. The snow depth data begins in 1955, and ends in 1994 for SW, 1997 for MJ, 2002 for LE, 2003 for WI, 2005 for RG and SK, 2006 for ES, GP, MH, PA, RD, RG, and TP, and 2010 for ED. This synthesis paper extracts significant results from many analyses [9–16], which use different subsets of the data, ranging from all station-years with snow depth (e.g., Section 3.1) to selected representative stations, which we will identify in the text.

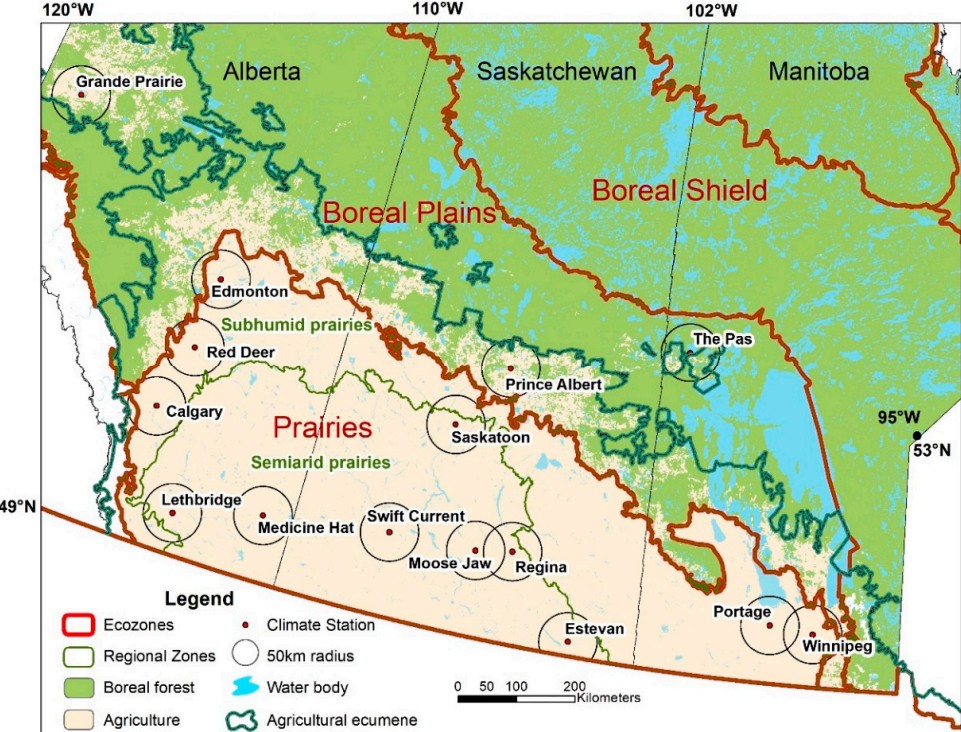

**Figure 1.** Climate station locations, Canadian ecozones, regional zones, agricultural regions, and boreal forest (adapted from [14]).

**Table 1.** Climate stations with locations and elevation.

| Station Name (Code) | Station ID | Province | Latitude | Longitude | Elevation (m) |
|---|---|---|---|---|---|
| Red Deer (RD) | 3025480 | Alberta | 52.18 | −113.62 | 905 |
| Calgary (CA) | 3031093 | Alberta | 51.11 | −114.02 | 1084 |
| Edmonton (ED) | 3012202 | Alberta | 53.57 | −113.52 | 671 |
| Lethbridge (LE) | 3033880 | Alberta | 49.63 | −112.80 | 929 |
| Medicine Hat (MH) | 3034480 | Alberta | 50.02 | −110.72 | 717 |
| Grande Prairie (GP) | 3072920 | Alberta | 55.18 | −118.89 | 669 |
| Regina (RG) | 4016560 | Saskatchewan | 50.43 | −104.67 | 578 |
| Moose Jaw (MJ) | 4015320 | Saskatchewan | 50.33 | −105.55 | 577 |
| Estevan (ES) | 4012400 | Saskatchewan | 49.22 | −102.97 | 581 |
| Swift Current (SW) | 4028040 | Saskatchewan | 50.3 | −107.68 | 817 |
| Prince Albert (PA) | 4056240 | Saskatchewan | 53.22 | −105.67 | 428 |
| Saskatoon (SK) | 4057120 | Saskatchewan | 52.17 | −106.72 | 504 |
| Portage-Southport (PS) | 5012320 | Manitoba | 49.9 | −98.27 | 270 |
| Winnipeg (WI) | 5023222 | Manitoba | 49.82 | −97.23 | 239 |
| The Pas (TP) | 5052880 | Manitoba | 53.97 | −101.1 | 270 |

## 2.2. Diurnal Range Definition

The diurnal range of temperature, DTR, is defined as the difference between the maximum temperature, $T_x$, and the minimum temperature, $T_n$:

$$DTR = T_x - T_n \tag{1a}$$

Similarly for relative humidity, RH, (and other variables), we define the diurnal range, DRH, as the difference between the maximum, $RH_x$, and the minimum, $RH_n$:

$$DRH = RH_x - RH_n \tag{1b}$$

In our early papers [9–13] we generally reduced the hourly data to daily means, $T_m$, $RH_m$, and recorded $T_x$, $T_n$, and DTR. The difference in relative humidity (RH), DRH, between $T_n$ and $T_x$ was used as an approximation of the diurnal range. However, there has been considerable discussion in recent years about the difference between DTR, $T_x$, and $T_n$ derived from the monthly means of hourly data, and the conventional monthly mean of daily values of DTR, $T_x$, and $T_n$ [12,17–19].

We explored this issue [14], using stratifications by month and by opaque cloud cover, and found systematic biases, especially in winter, and even in summer under cloudy conditions. We concluded that the radiatively-forced diurnal cycle, that is, the lagged response to the diurnally varying radiation field, which is dependent on opaque cloud cover, is represented best by first binning the hourly data for groups of many days, and then by determining the diurnal ranges from the composites. Specifically, we found that this radiatively-forced diurnal cycle has a smaller amplitude than the corresponding average of the daily ranges. The reason is transparent. Without an advection of temperature, $T_n$ is near sunrise and $T_x$ is in the mid-afternoon, but advection can shift the daily minimum temperature away from the time of sunrise to a lower value than the temperature at sunrise, and similarly, advection can shift the daily maximum temperature away from the mid-afternoon to a higher value than the mid-afternoon temperature. Either will give a larger diurnal range.

Our dataset has around 240,000 days, so coarse stratifications may have 2000 days in each bin, and detailed sub-stratifications typically have >200 days in each bin. This means that the radiatively-forced diurnal cycle emerges from composites of the hourly data, since the advection of temperature and humidity varies from day to day. This leads to a fundamental quantitative improvement in our understanding of the coupling between the diurnal cycle and the opaque cloud cover that determines the cloud radiative forcing.

We also derived from T, RH, and surface pressure, PS, the other thermodynamic variables: the mixing ratio (Q), the potential temperature ($\theta$), the equivalent potential temperature ($\theta_E$), and the saturation pressure (p*) at the lifting condensation level (LCL). We defined the pressure height to the

LCL, $P_{LCL}$ = PS − p* [2], which in the warm season, is often an indicator of the height of cloud base [9]. We calculated the diurnal ranges that are related to moist convective processes:

$$D\theta_E = \theta_{Ex} - \theta_{En} \tag{2a}$$

$$DP_{LCL} = P_{LCLx} - P_{LCLn} \tag{2b}$$

### 2.3. Opaque Cloud Bins

Since opaque cloud reflects the solar flux and traps the outgoing long-wave, we used the daily mean of the hourly opaque cloud measurements to stratify the daily mean data, and the diurnal ranges of temperature and humidity and derived variables (see Section 3). In fact, we computed two daily averages from the all-sky opaque cloud cover estimates to use for stratification. The first is the simple mean of the 24 hourly values, $OPAQ_m$. The second, OPAQSW, is a mean of the hourly opaque cloud values during daylight hours, weighted by a fit to the downward clear sky flux derived from the reanalysis known as ERA-Interim (details in [13]).

### 2.4. Cloud Radiative Forcing

In the short-wave radiation budget, we can define an effective cloud albedo (ECA) and the short-wave cloud forcing (SWCF) in terms of a downwelling SW clear-sky flux, $SWCS_{dn}$, based on a fit to the clear-sky fluxes from the nearest grid-point of the reanalysis ERA-Interim [13,20]:

$$ECA = 1 - SW_{dn}/SWCS_{dn} \tag{3}$$

$$SWCF = SWCS_{dn} - SW_{dn} = -ECA * SWCS_{dn} \tag{4}$$

The dimensionless ECA, with a range from 0 to 1, is a useful measure of the impact of the reflective cloud field on the surface shortwave radiation budget [2,3]. SWCF becomes increasingly negative as ECA increases, while $SWCS_{dn}$ has a large increase from the winter to the summer solstice.

Similarly, we can define a long-wave cloud forcing (LWCF) in terms of a downwelling clear-sky flux $LWCS_{dn}$, also from ERA-Interim, as:

$$LWCF = LW_{dn} - LWCS_{dn} \tag{5}$$

$LWCS_{dn}$ is the smaller term, and $LW_{dn}$ increases with increasing cloud cover, so that LWCF is positive.

The total cloud forcing (CF) of the downwelling radiative fluxes is the sum:

$$CF = SWCF + LWCF \tag{6a}$$

In the warm season, the SWCF dominates, and CF is negative. The net cloud forcing can be defined as:

$$CF_{net} = (1 - \alpha_s) SWCF + LWCF \tag{6b}$$

where the mean surface albedo, defined as:

$$\alpha_s = SW_{dn}/SW_{up} \tag{7}$$

ranges for Saskatchewan from about 0.18 in summer to 0.73 in winter with snow cover [11,12]. When there is a snow cover, the positive LWCF dominates, because the lower solar elevation and larger surface albedo greatly reduce the net SWCF.

We computed the net LW flux:

$$LW_n = LW_{dn} - LW_{up} \tag{8}$$

using observations for $LW_{dn}$, and estimating $LW_{up}$ from the daily mean air temperature, $T_m$ (°C), from:

$$LW_{up} = \varepsilon \, \sigma \, T_k{}^4 \tag{9}$$

with $T_k$ (K) = $T_m$ + 273.15, $\sigma = 5.67 \times 10^{-8}$ (W m$^{-2}$ K$^{-4}$) and the emissivity $\varepsilon$ set to 1.

*2.5. Data*

Data Availability: The Canadian Prairie data are available from the first author, or from Environment and Climate Change Canada at http://climate.weather.gc.ca/. The reanalysis data are available from ECMWF at https://ecmwf.int/en/research/climate-reanalysis/era-interim.

## 3. Climate Coupling to Opaque Cloud and Snow Cover

This section will present several topics: the monthly diurnal cycle with and without snow cover, the relationship between snow cover, opaque cloud, and cloud radiative forcing, the climate impact of snow cover, the coupling between opaque cloud and warm season diurnal thermodynamic ranges, and the dependence of the 24 h imbalances of the diurnal cycle on opaque cloud cover.

*3.1. Forcing of Diurnal Cycle by Cloud and Snow Cover*

We start with the dependence of the monthly diurnal cycle of temperature on cloud and snow cover [14]. Taking the data from all stations-years in Table 1 that have snow depth data, we first stratified by temperature and snow cover: selecting the warm group of days with $T_m > 0$ °C and no snow cover (141,160 days), and the cold group of days with $T_m < 0$°C with surface snow cover (74,260 days). Here, we exclude the much smaller mixed group of days, above freezing with snow cover and below freezing without snow (see [14]).

Figure 2 shows the mean diurnal cycle of temperature by month, stratified into 10 bins of daily mean opaque cloud, $OPAQ_m$. In the warm season from May to October, we see a steep increase of maximum temperature $T_x$ and diurnal temperature range DTR with decreasing opaque cloud, and a rather small fall of minimum temperature in summer. The changing day-length is clearly visible by September and October. In sharp contrast, in the cold season with snow, from December to February, $T_x$ decreases with decreasing opaque cloud, and $T_n$ decreases even more steeply to its lowest minimum at sunrise under clear skies. Beside the September and December plots, we show $OPAQ_m$ legends in ascending and descending order to illustrate this reversal of the diurnal cycle coupling to opaque cloud between warm and cold seasons.

For the transition months, November, March, and April between warm and cold seasons, both regimes with and without snow cover are well-represented; it is clear that the distributions are non-overlapping. Note that the temperature range shown for the transition months is broader (32 K) than for the single months with a single regime (21 K). This is a large dataset with about 20,000 days per month, so that each cloud bin has typically about 2000 days in summer and winter. For the transition months, where the data is also split unevenly, the number in each bin varies from about 200 to 1500.

It is clear that snow cover has two large climate impacts. First, it cools the mean climate, represented by $T_m$, by about 10 °C; and second, it reverses the sign of the coupling to opaque cloud cover. Snow cover acts as a climate switch between non-overlapping regimes [11,14]. We will explore the climate impact of snow cover further in Section 3.4, but first we will show the seasonal impact on the cloud radiative forcing.

*3.2. Change of Cloud Forcing between Warn and Cold Season*

The dramatic differences in the diurnal cycles of temperature shown in Figure 2 are related to the reversal of the sign of the net cloud forcing between the warm season and the cold season with snow cover. We computed this using Equations (3)–(7), and data from the Baseline Surface Radiation Network (BSRN) Prairie site at Bratt's Lake, Saskatchewan at 50.204° N, 104.713° W, elevation 588 m [13]. We

have 17 years of the downwelling fluxes, $SW_{dn}$ and $LW_{dn}$, at Bratt's Lake, which we first averaged from 1-min data to hourly means, and then to daily means.

Figure 3 shows that $CF_{net}$ from Equation (6b) reverses the sign from increasing negative with cloud cover in the warm season, to increasing positive in the cold season with cloud cover. This is consistent with the daily mean temperature response seen in Figure 2 to the changing opaque cloud cover.

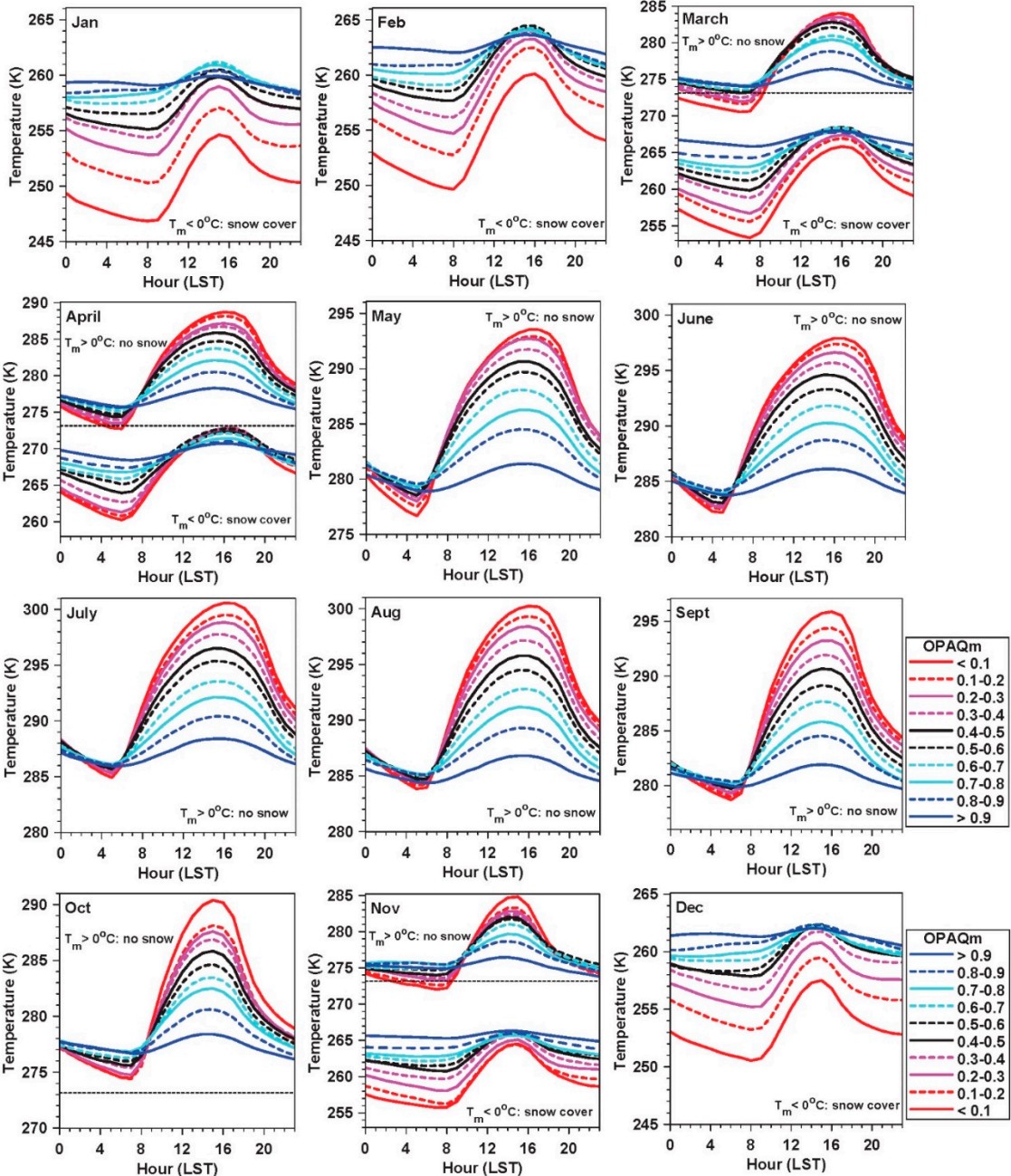

**Figure 2.** Monthly diurnal cycles for cold-snow and warm-no-snow classes, stratified by opaque cloud (adapted from [14]).

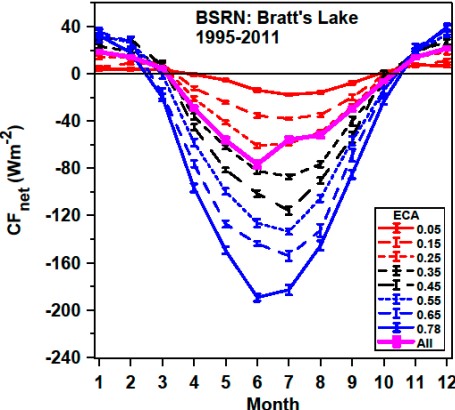

**Figure 3.** Mean annual cycle of $CF_{net}$, stratified by effective cloud albedo (ECA) (adapted from [13]).

### 3.3. Relationship between Opaque Cloud and Cloud Radiative Forcing

We then binned the BSRN data from Bratt's Lake for the downward SW and LW fluxes using the opaque cloud measurements at Regina, 25 km to the north, simply defining the warm season as days with $T_m > 0\ °C$ and the cold season as days with $T_m < 0\ °C$, because we have no snow cover data for Bratt's Lake. For the SW comparison, we compared the daytime weighted opaque cloud, OPAQSW (see Section 2.3) with ECA from Equation (3). For the LW comparison, we compared the 24 h mean $OPAQ_m$ with $LW_n$ computed from Equation (8).

Figure 4 (left) shows the relationship between ECA and OPAQSW for the warm season above freezing, and the cold season below freezing. ECA increases more steeply with increasing opaque cloud in the warm season than in the cold season. We show the mean and standard error of the binned data, and quadratic regression fits to the daily data, which could be used to convert opaque cloud to ECA. For the warm season, the fit is ($R^2 = 0.87$):

$$ECA = 0.06(\pm 0.08) + 0.02(\pm 0.02)\ OPAQSW + 0.65(\pm 0.02)\ OPAQSW^2 \tag{10a}$$

For the cold season, the fit is ($R^2 = 0.71$):

$$ECA = 0.07(\pm 0.11) + 0.08(\pm 0.03)\ OPAQSW + 0.37(\pm 0.03)\ OPAQSW^2 \tag{10b}$$

The uncertainty in ECA on a daily basis is of the order of $\pm 0.08$ in the warm season and $\pm 0.11$ in the cold season. The standard errors (SE) shown for the climatological fits are much smaller, because they are reduced by the large number of days.

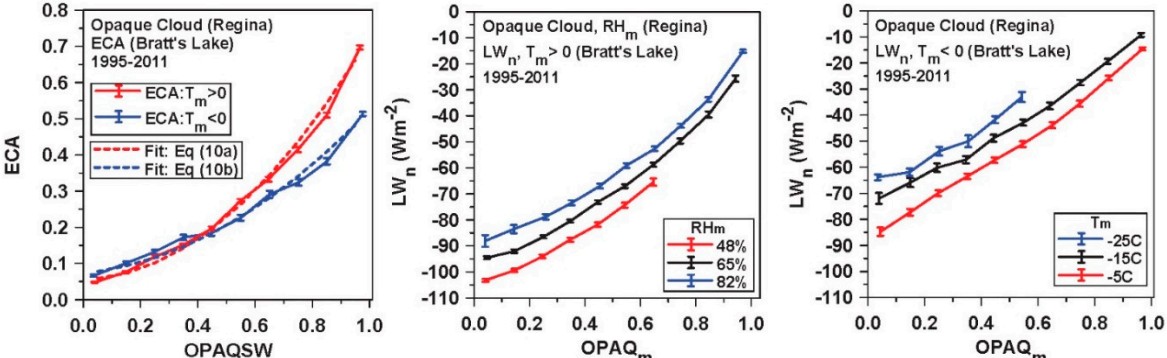

**Figure 4.** Relationship between opaque cloud at Regina and Bratt's Lake ECA (**left**), opaque cloud, and $LW_n$ stratified by $RH_m$ in the warm season (**middle**), and (**right**) $LW_n$ stratified by $T_m$ in the cold season (adapted from [13]).

Figure 4 (middle) shows the dependence of $LW_n$ on opaque cloud for days above freezing (3245 days) for three bins of daily mean $RH_m$ (<60, 60–75, >75%). The outgoing $LW_n$ flux for the same cloud cover increases as RH falls. The temperature dependence is very small when $T_m > 0°C$ (not shown). The right panel shows the dependence of $LW_n$ on opaque cloud for temperatures below freezing (2198 days) for three bins of daily mean $T_m$ (<−20, −20 to −10, −10 to 0 °C). The outgoing $LW_n$ flux now decreases with colder temperatures, probably because the surface cools under a stable BL in the cold season [13].

In the warm season, multiple regression of the daily values of $LW_n$ on quadratic opaque cloud and $RH_m$ gives ($R^2 = 0.91$):

$$LW_n = -128.6(\pm 7.8) + 28.1(\pm 1.8)OPAQ_m + 44.6(\pm 1.8)OPAQ_m{}^2 + 0.49(\pm 0.01)RH_m \tag{11a}$$

In the cold season, multiple regression on quadratic opaque cloud, $T_m$ and $RH_m$ gives ($R^2 = 0.83$):

$$LW_n = -112.2(\pm 9.8) + 43.5(\pm 2.8)OPAQ_m + 26.8(\pm 2.5)OPAQ_m{}^2 +$$
$$0.29(\pm 0.02)RH_m - 1.02(\pm 0.03)T_m \tag{11b}$$

### 3.4. Climate Impact of Snow Cover

Figure 2 shows that the impact of snow cover on the Prairies on the diurnal cycle of temperature is very large. The transition months show that the cooling with snow cover is large, and show a reversal of the response to cloud cover, consistent with the reversal of the net cloud forcing between cold and warm seasons shown in Figure 3. This section addresses the resulting mean climate impact of snow cover.

Figure 5 shows four different analyses of the climate impact of snow cover. The top-left is a composite of the six climate stations in Saskatchewan for eight days before and after fresh snowfall in November, showing a mean of about 270 snowfall events, with a mean date of November 15 (adapted from [11]). We see the fall of daily mean temperature across the snow event, from near 0 °C a week before, to −9.4 ± 0.7 °C for days 2 to 8 afterwards. The climate transition from fall to winter often comes abruptly with these snow events [11], as the snowpack may not melt till spring. Similar composites for individual stations and the means for other provinces are shown in [11]. All of these suggest that as the albedo of the Prairies changes from about 0.2 with no snow cover to above 0.7 with snow cover [11,12], there is a fall of temperature of nearly 10 °C, and the reverse change occurs in spring with snow melt (see [11]).

The top-right (adapted from [14]) shows the fall of mean daily temperature, $\delta T_m$, with snow cover, derived from Figure 2 by calculating the difference of the diurnal composites with and without snow for the transition months, November and March, for each opaque cloud cover bin. We made a correction of about 2 °C, based on the mean seasonal cycle [14], to allow for the fact that the mean date of the snow-free composite is about 15 days earlier in November, and later in March than the composite with snow. The curves are a little noisy, because the independent sampling in opaque cloud bins, with and without snow, is far from homogeneous, and in these transition months, the number of days in each bin ranges widely from 184 to 1869 (not shown). Nonetheless, we see a larger degree of cooling as the opaque cloud decreases. The climate cooling with snow, averaged across all cloud bins (open circles), is −11.8 °C (−10.7 °C) for November (March). We also show quadratic fits (dashed) as a useful smooth reference for the impact of cloud cover. We note that the radiative forcing is stronger in March than November, but we cannot assess whether the small difference between the November and March curves is significant, given the inhomogeneity across the cloud bins.

The bottom-left plot shows the monthly mean temperature across the cold season (black line) and the partition into days with snow cover (blue) and days with no snow cover (red line) for a single station (Lethbridge, Alberta), together with the mean snow depth. The difference between the blue and red curves (the magenta curve) shows the monthly climate cooling of snow cover with a mean value

of $\Delta T = -10.4 \pm 0.4$ °C. The standard errors shown are small because of the large number of days in the 49-year record. Other stations show similar plots [15], suggesting that the cold season climatology with and without snow (red and blue curves) are distinct and non-overlapping. Conventionally, they are merged to the black curve, so this can be misleading.

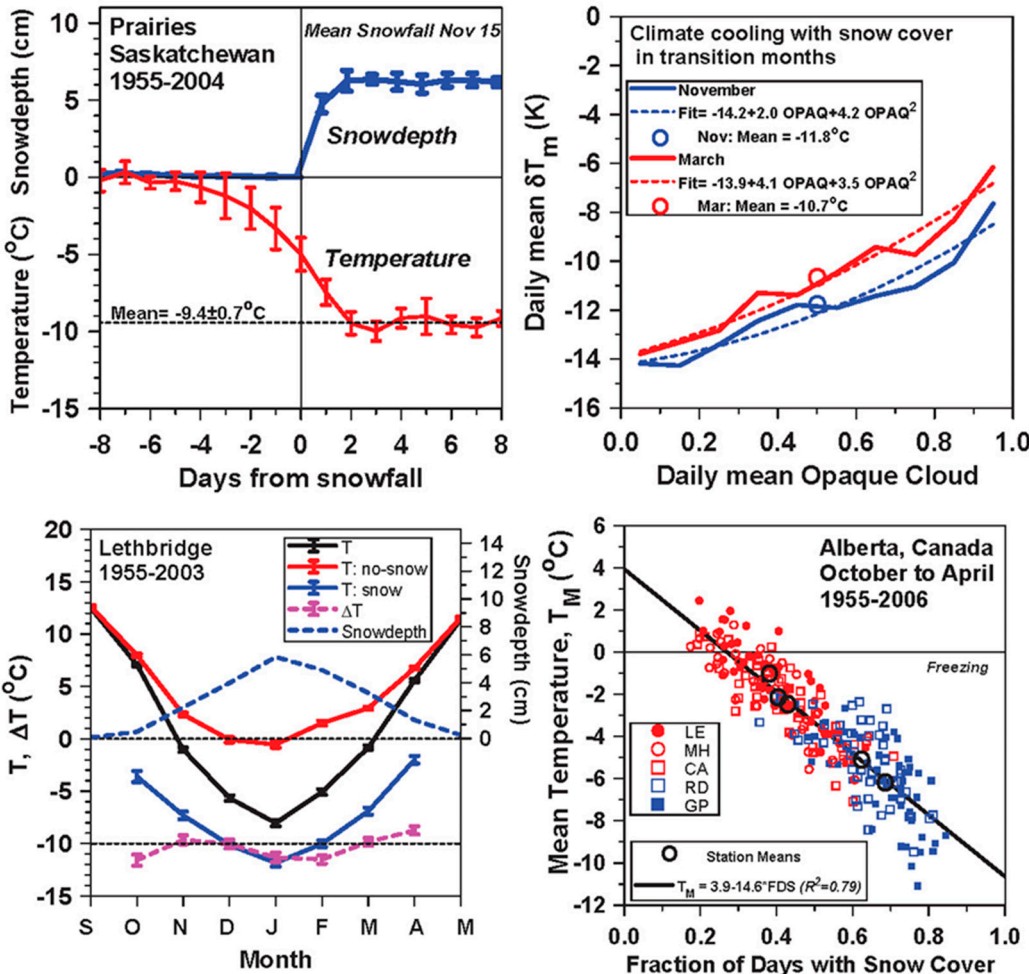

**Figure 5.** Drop of temperature with fresh snowfall (**top-left**), climate cooling with snow cover in November and March as a function of opaque cloud (**top-right**), 10 °C separation of cold season climates with and without snow cover (**bottom-left**), and (**bottom-right**) dependence of mean cold season temperature on fraction of days with snow cover (adapted from [11,14,15].

The bottom-right panel shows the mean temperature $T_M$, for October to April against the fraction of days with snow cover (FDS) for five stations in Alberta, Lethbridge, Medicine Hat, Calgary, Red Deer, and Grande Prairie, listed in order of increasing latitude (adapted from [11,15]). The line fit shown is for 326 years of data, and we show the station means (black circles) that lie close to this line fit. Since it is clear that the southern three stations (red points) have warmer temperatures and lower FDS than the northern two stations, we also computed the linear regression slopes for these two groups.

$$\text{All station fit} \quad T_M = 3.9(\pm 1.2) - 14.6(\pm 0.5) * \text{FDS} \quad (R^2 = 0.79) \tag{12a}$$

$$\text{3 southern station fit} \quad T_M = 3.8(\pm 1.5) - 14.3(\pm 0.7) * \text{FDS} \quad (R^2 = 0.73) \tag{12b}$$

$$\text{2 northern station fit} \quad T_M = 3.2(\pm 1.5) - 13.6(\pm 1.5) * \text{FDS} \quad (R^2 = 0.48) \tag{12c}$$

These agree within the uncertainty, which increases for fewer stations. The corresponding plot for Saskatchewan is similar [11]. We conclude that the climate coupling between the fraction of days with snow cover and the mean cool season temperature is a robust feature of the Prairie landscape. The shift of the station means with increasing latitude suggests that reduced insolation is also playing a tightly coupled role.

Figure 5 confirms that snow cover has a large cooling impact on the mean temperatures in the cold season: snow cover acts as a climate switch between the two non-overlapping regimes. On daily timescales, the cooling is about $-10\,^\circ$C for the Prairies, where the surface albedo with snow cover is in the order of 0.7. The larger slope of $-14.6\,^\circ$C in fit (12a) for the change of mean cold season temperature with the fraction of days with snow cover suggests that there may be coupling to larger scales that enhance the regional cooling with snow cover.

### 3.5. Coupling of Warm Season Diurnal Ranges and 24-h Imbalances to Opaque Cloud

This very large hourly dataset allowed us for the first time to extract the radiatively forced diurnal ranges, shown in Equations (1) and (2), for the key thermodynamic variables [14]. Here, we will just show the warm season; the cold season can be found in [14]. From Figure 2, we extracted DTR as a function of opaque cloud and month, and we extracted DRH, $D\theta_E$, and $DP_{LCL}$ from similar diurnal composites (not shown). Close examination of Figure 2 shows that there is a discontinuity across local midnight that changes with opaque cloud cover. So we calculated, also for the first time, this 24 h imbalance of the diurnal cycle as a function of opaque cloud and month. These are key conceptual improvements in our understanding of the diurnal cycle over land in the warm season, and our results are robust as there are about 20,000 days per month.

Figure 6 (top left panel) shows the mean diurnal ranges of temperature, DTR, relative humidity, DRH, and mean daily precipitation for the warm season months April to September with no snow. Remarkably, the diurnal ranges are tightly clustered [9,14], so we also show the 6-month warm season mean. The quadratic regression fits for the dependence of the 6-month mean DTR and DRH on $OPAQ_m$ are:

$$DTR = 16.7(\pm 0.4) - 9.3(\pm 0.8) * OPAQ_m - 6.0(\pm 0.7) * OPAQ_m{}^2 \quad (R^2 = 0.992) \tag{13a}$$

$$DRH = 47.5(\pm 0.8) - 2.6(\pm 1.4) * OPAQ_m - 38.9(\pm 1.4) * OPAQ_m{}^2 \quad (R^2 = 0.996) \tag{13b}$$

The leading coefficient is the clear-sky diurnal range, which is a rise of $16.7\,^\circ$C to the afternoon maximum, coupled to a fall of 47.5% in RH from the morning maximum at sunrise. The cloudy limit for $OPAQ_m = 1$, given by these fits, are the small values (DTR, DRH) = ($1.4\,^\circ$C, 6.0%).

Monthly mean precipitation is very low for $OPAQ_m < 0.4$, and the increase of precipitation with $OPAQ_m$ is largest in summer, peaking in July when T and the mixing ratio Q also peak. However, because June has substantially greater opaque cloud cover [12], mean June precipitation (2.28 mm d$^{-1}$) is greater than July (1.91 mm d$^{-1}$).

Figure 6 (top right) shows the 24 h imbalances of $\Delta T_{24}$ and $\Delta RH_{24}$, which we calculated from the discontinuities across local midnight [14]. We see that over the range of $OPAQ_m$ from 0.05 to 0.95 (nearly clear to nearly opaque cloud cover), the mean ($\Delta T_{24}$, $\Delta RH_{24}$) change monotonically from (+2 $^\circ$C, $-6$%) to ($-1.5\,^\circ$C, +6%). Under nearly clear skies, the warming, and drying over the diurnal cycle is slightly larger in April, May, and June when the mean temperature is increasing seasonally, and slightly smaller in August and September. Under cloudy skies, there is a larger increase in $\Delta RH_{24}$ in April and May. The SE of the hourly binned data from which Figure 6 is derived as $\approx 0.1$ K for T, $\leq 0.5$% for RH.

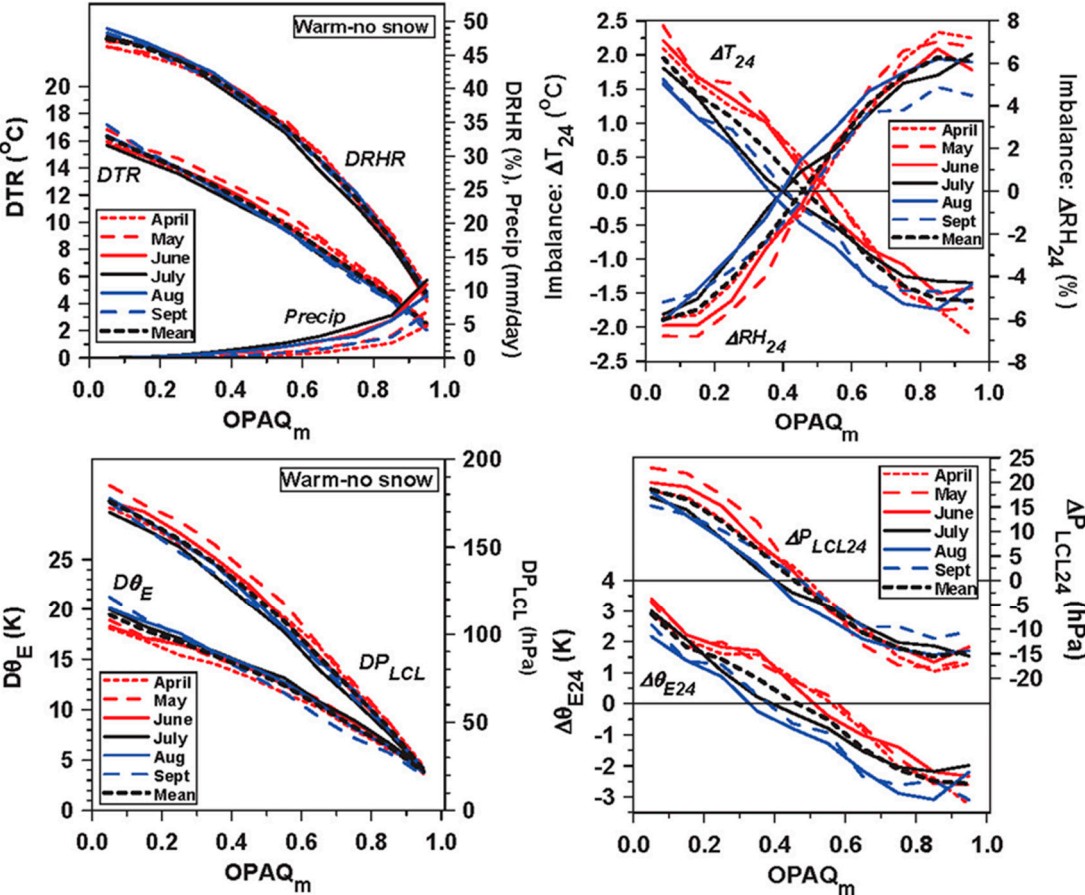

**Figure 6.** The opaque cloud dependence of the diurnal ranges of T, RH, $\theta_E$, and $P_{LCL}$ (**left**) and (**right**) the 24 h imbalance of the diurnal cycle (adapted from [14]).

The warming of +2 °C and a drying of −6% over the diurnal cycle under nearly clear skies is about 12% of both the DTR and DRH. The cooling of −1.5 °C and a moistening of +6% under very cloudy skies may be coupled to both the evaporation of rain and downdraft transports. The uniform progression of the diurnal imbalance with increasing cloud is not surprising. However, this means that a steady state diurnal cycle only exists under partly cloudy conditions: for the 6-month mean, $\Delta T_{24}$, and $\Delta RH_{24}$ cross zero for $OPAQ_m = 0.45$. This presents a conceptual challenge for equilibrium models for the non-precipitating convective BL over land [21].

Figure 6 (lower panels) are the corresponding warm season diurnal ranges and 24 h imbalances for $\theta_E$ and $P_{LCL}$. The spread in the diurnal ranges and the diurnal imbalances is again small from April to September. For the 6-month means, the quadratic regression fits for the $OPAQ_m$ dependence are:

$$D\theta_E = 19.7(\pm0.7) - 9.4(\pm1.2) * OPAQ_m - 7.5(\pm1.2) * OPAQ_m^2 \ (R^2 = 0.983) \tag{14a}$$

$$DP_{LCL} = 181.4(\pm4.9) - 90.3(\pm9.0) * OPAQ_m - 81.1(\pm8.7) * OPAQ_m^2 \ (R^2 = 0.991) \tag{14b}$$

Again, the leading coefficient is the clear-sky diurnal range, which is a rise of ($\theta_E$, $P_{LCL}$) = (19.7 K, 181.4 hPa) from the morning sunrise minimums. The cloudy limit for $OPAQ_m = 1$, given by these fits, are the small values ($D\theta_E$, $DP_{LCL}$) = (2.73 K, 10.0 hPa).

The bottom-right panel for the corresponding monthly mean 24 h imbalances shows that under nearly-clear skies, there is an increase of +2.9 K for $\Delta\theta_{E24}$ and +18.6 hPa for $\Delta P_{LCL24}$, which are 14.9% and 10.5% of the respective diurnal ranges. There is a corresponding small 24 h imbalance of mixing ratio, $\Delta Q_{24}$, of +0.2 g kg$^{-1}$ (not shown). At the other limit under nearly-overcast skies, typically with rain, the 24 h imbalance is a fall of −2.6 K for $\Delta\theta_{E24}$ and −14.6 hPa for $\Delta P_{LCL24}$, with a corresponding

fall of $\Delta Q_{24}$ of $-0.24$ g kg$^{-1}$ (not shown). The SE of the hourly binned data from which these plots are derived is $\leq 0.3$ K for $\theta_E$ and $\leq 1.5$ hPa for $P_{LCL}$. On the seasonal timescale, we see that the imbalance of $\Delta\theta_{E24}$ is larger in April, May, and June over most of the OPAQ$_m$ range as the climate warms, and smaller in August and September. However for $\Delta P_{LCL24}$, the seasonal response has an asymmetric structure that is consistent with $\Delta RH_{24}$, since a lower RH is tightly coupled to a higher $P_{LCL}$.

Figure 6 shows the remarkably tight climatological coupling from April to September which links opaque cloud cover, the diurnal ranges and the 24 h diurnal imbalances, despite substantial differences in the solar zenith angle. Over the diurnal cycle, under nearly clear skies, we see a warming and drying, and a rise of $\theta_E$ and $P_{LCL}$. At the cloudy extreme with rain, we see 24 h imbalances of opposite sign that are generally slightly smaller. These rather precise warm-season patterns across opaque cloud cover, and therefore cloud radiative forcing, set a clear target for modeling the partially cloudy boundary layer over land.

## 4. Hydrometeorological Memory on Monthly Timescales

The close coupling between the energy and water cycles at the land surface is central to hydrometeorology, and important to weather forecasts on timescales from days to seasons. An earlier review looked at hydrometeorology using global model reanalysis data [1], which showed how net long-wave and short-wave radiation, cloud cover, surface fluxes, diurnal temperature range, soil moisture, and cloud-base height were coupled on daily timescales over river basins. Reanalysis data contain all the key variables, but historically, the observed near-surface climate variables were temperature and precipitation, along with pressure, wind, relative humidity, and snow-depth. Section 3.5 shows that the warm season diurnal cycle is dominated by the radiative forcing of the opaque cloud cover. But on monthly and longer timescales, soil moisture anomalies are linked to precipitation anomalies, both for the current month and several preceding months.

Here, we summarize some key results from [16], who merged the 12 stations in Table 1 in Alberta and Saskatchewan for the years when precipitation is available. For this monthly analysis, the hourly data were processed as intact monthly mean diurnal cycles for each station for each year. As noted in Section 2.1, the hourly dataset is remarkably complete. Days were omitted if <20 h of data were available. Months were omitted if they had fewer than 28 days remaining, except for February, where this threshold was reduced to 25 days. From the monthly diurnal cycles of T, RH, and PS, we computed the derived thermodynamic variables, Q, $\theta_E$, and $P_{LCL}$, and the diurnal ranges defined in Equations (1) and (2).

For each variable, Y, we extracted from the monthly mean diurnal cycles, the daily mean, $Y_m$, the maximum and minimum, $Y_x$ and $Y_n$, and the times of the maximum and minimum [16]. We then computed the long-term station monthly mean, and used these to compute monthly anomalies, $\delta Y$. For the daily precipitation and snow-depth, we also computed monthly means, the long-term station monthly means, and used these to compute monthly anomalies for each station. The monthly anomalies of opaque cloud, precipitation, snow depth, and snow cover frequency were then standardized by their monthly standard deviation (SD). For the temperature anomalies, $\delta T_m$, $\delta T_x$, $\delta T_n$, and the diurnal temperature range, $\delta DTR$, we standardized by the monthly SD of $\delta T_m$. Similarly for the variables, $\delta RH_m$, $\delta RH_x$, $\delta RH_n$ and the diurnal RH range $\delta DRH$, we standardized by the monthly SD of $\delta RH_m$. The corresponding set of anomalies for equivalent potential temperature, $\delta\theta_E$, and pressure–height to the LCL, $\delta P_{LCL}$, were standardized by the monthly SD of $\delta\theta_{Em}$, and $\delta P_{LCLm}$ respectively.

We used multiple linear regression to explore the correlation between variables. Following [12,16], our starting format was to regress a standardized thermodynamic anomaly, $\delta Y$, on opaque cloud anomalies ($\delta OPAQ_m$) for the current month, and lagged precipitation anomalies for the current month ($\delta PR0$) and preceding months ($\delta PR1$, $\delta PR2$, $\delta PR3$, $\delta PR4$, $\delta PR5$) in the form:

$$\delta Y = A * \delta OPAQ + B * \delta PR0 + C * \delta PR1 + D * \delta PR2 + E * \delta PR3 + F * \delta PR4 + G * \delta PR5 \qquad (15)$$

Multiple regression shows no memory of cloud for previous months. Since we are using anomalies, the leading coefficient is of order zero, so it is not shown. After standardization, all variables are dimensionless.

### 4.1. Memory of Cold Season Precipitation in April Climatology

On the Prairies, precipitation memory lasts through winter, as water is stored until the snowpack melts in late March or April. The reflective snow cover on the Prairies, with an albedo ≈0.7, acts as a climate switch that reduces $T_m$ by 10 °C (Figures 2 and 5). April is the month when the snowpack finally melts and the ground thaws. The upper group in Table 2 shows selected April standardized anomalies regressed on standardized anomalies of opaque cloud for April; and precipitation from April back to November (coefficients A to G in Equation (15)). We see that the April monthly anomalies show memories of the anomalies of precipitation 5 months back through the entire cold season to November, when typically the ground begins to freeze, and the first lasting snow occurs (Figure 5). Some of this memory remains in the March snowpack depth (not shown here, see [16]).

**Table 2.** Standardized regression coefficients for April anomalies in anomalies $\delta$DTR (diurnal range of temperature), $\delta T_x$, $\delta RH_n$, $\delta RH_m$, and $\delta P_{LCLx}$ on standardized anomalies of opaque cloud and precipitation (upper group); and (lower group) adding fraction of April days with snow cover. For coefficients: plain text represents $p < 0.01$ (>99%); italic represents $0.01 \leq p < 0.05$, and coefficients are omitted for $p > 0.05$.

| Variable 620 months $R^2$ | $\delta$DTR 0.67 | $\delta T_x$ 0.47 | $\delta RH_n$ 0.65 | $\delta RH_m$ 0.63 | $\delta P_{LCLx}$ 0.66 |
|---|---|---|---|---|---|
| $\delta OPAQ_m$-Apr (A) | $-0.52 \pm 0.02$ | $-0.78 \pm 0.04$ | $0.76 \pm 0.03$ | $0.60 \pm 0.03$ | $-0.93 \pm 0.04$ |
| $\delta$PR-Apr (B) | $-0.06 \pm 0.02$ | | $0.20 \pm 0.03$ | $0.17 \pm 0.03$ | $-0.19 \pm 0.04$ |
| $\delta$PR-Mar (C) | $-0.12 \pm 0.02$ | $-0.22 \pm 0.04$ | $0.23 \pm 0.03$ | $0.19 \pm 0.02$ | $-0.27 \pm 0.03$ |
| $\delta$PR-Feb (D) | $-0.07 \pm 0.02$ | $-0.12 \pm 0.04$ | $0.16 \pm 0.03$ | $0.13 \pm 0.02$ | $-0.19 \pm 0.03$ |
| $\delta$PR-Jan (E) | $-0.09 \pm 0.02$ | $-0.19 \pm 0.04$ | $0.17 \pm 0.03$ | $0.13 \pm 0.02$ | $-0.21 \pm 0.03$ |
| $\delta$PR-Dec (F) | $-0.06 \pm 0.02$ | | $0.16 \pm 0.03$ | $0.14 \pm 0.02$ | $-0.19 \pm 0.03$ |
| $\delta$PR-Nov (G) | $-0.08 \pm 0.02$ | $-0.13 \pm 0.04$ | *$0.07 \pm 0.03$* | $0.08 \pm 0.02$ | $-0.11 \pm 0.03$ |
| **Variable 550 months $R^2$** | **$\delta$DTR 0.73** | **$\delta T_x$ 0.65** | **$\delta RH_n$ 0.80** | **$\delta RH_m$ 0.70** | **$\delta P_{LCLx}$ 0.78** |
| $\delta OPAQ_m$-Apr (A) | $-0.49 \pm 0.02$ | $-0.57 \pm 0.04$ | $0.65 \pm 0.03$ | $0.54 \pm 0.03$ | $-0.82 \pm 0.04$ |
| $\delta$PR-Apr (B) | *$-0.04 \pm 0.02$* | | $0.16 \pm 0.03$ | $0.15 \pm 0.03$ | $-0.15 \pm 0.04$ |
| $\delta$PR-Mar (C) | $-0.08 \pm 0.02$ | *$-0.07 \pm 0.03$* | $0.14 \pm 0.03$ | $0.14 \pm 0.03$ | $-0.18 \pm 0.03$ |
| $\delta$PR-Feb (D) | $-0.05 \pm 0.02$ | | $0.09 \pm 0.03$ | $0.10 \pm 0.03$ | $-0.11 \pm 0.03$ |
| $\delta$PR-Jan (E) | $-0.05 \pm 0.02$ | | *$0.06 \pm 0.03$* | $0.07 \pm 0.03$ | $-0.08 \pm 0.03$ |
| $\delta$PR-Dec (F) | $-0.04 \pm 0.02$ | | $0.12 \pm 0.02$ | $0.13 \pm 0.02$ | $-0.16 \pm 0.03$ |
| $\delta$PR-Nov (G) | $-0.06 \pm 0.02$ | $-0.10 \pm 0.03$ | | | |
| $\delta$SnowCover-Apr (S) | $-0.19 \pm 0.02$ | $-0.63 \pm 0.04$ | $0.52 \pm 0.03$ | $0.31 \pm 0.03$ | $-0.57 \pm 0.03$ |

For the first row, $\delta$OPAQ-Apr, the large negative coefficients for the monthly anomalies $\delta$DTR, $\delta T_x$, and $\delta P_{LCLx}$, mean that these variables decrease with increasing opaque cloud cover, while the positive sign for the $\delta RH_n$ and $\delta RH_m$ means that they increase with opaque cloud. For $\delta T_x$ and $\delta$DTR (and $\delta T_m$, not shown), the negative coefficients B to G, for the months March back to November, mean that the positive cold season precipitation anomalies are correlated with cold April temperatures. For $\delta RH_n$, $\delta RH_m$ (and $\delta RH_x$, not shown), the positive coefficients, B to G, mean that positive cold season precipitation anomalies are correlated with higher RH in April. Most coefficients for $\delta$DTR, $\delta RH_n$, $\delta RH_m$, and $\delta P_{LCLx}$ (representative of afternoon cloud-base) have a 99% confidence ($p < 0.01$).

There are several physical processes that are probably involved. The precipitation over the cold season is mostly stored in the snowpack till spring, when the melt absorbs energy and cools the surface; the melt also provides water for evaporation, which also cools and increases RH. In addition, the freeze-up of the soil in November may similarly preserve November precipitation anomalies as soil ice through the cold season until spring melt.

In April, the high albedo of the remaining snowpack, as well as fresh snow, also play a direct climate role, as discussed in Section 3.4 and shown in Figure 2, because snow cover acts as a climate switch. Thus, we computed the standardized April snow cover frequency anomaly from the fraction of days in April with snow depth >0, and added this to the multiple regression (15) to give:

$$\delta Y\text{-Apr} = A * \delta OPAQ_m\text{-Apr} + B * \delta PR\text{-Apr} + C * \delta PR\text{-Mar} + D * \delta PR\text{-Feb} +$$
$$E * \delta PR\text{-Jan} + F * \delta PR\text{-Dec} + G * \delta PR\text{-Nov} + S * \delta SnowCover\text{-Apr} \tag{16}$$

The lower group in Table 2 shows the coefficients from Equation (16). There is an increase in $R^2$ for all variables, and especially for $T_x$, with the addition of snow cover. For maximum temperature, $T_x$, snow cover frequency anomalies have as large an impact as opaque cloud anomalies. Note that the coefficients G for $\delta PR\text{-Nov}$ for $\delta RH_n$, $\delta RH_m$, and $\delta P_{LCLx}$ are not significant, but the coefficients G for $\delta DTR$ and $\delta T_x$ have a confidence >99% in Table 2. It is possible that this is the cooling impact in April coming from the melt of soil–ice that was frozen back in November.

Table 2 shows that the climate in April, when the snow pack finally melts, has memory of precipitation through the entire previous winter up to November. Most of the variability in the April climate is explained by anomalies of winter precipitation and the fraction of days in April with residual snow cover.

### 4.2. Growing Season Memory of Precipitation

After snowmelt on the Prairies, the transition into the growing season months May to August (MJJA) is rapid, and typically, the memory of precipitation for the months May to August only goes back to March or April [16]. Merging the 2466 MJJA growing season months gives a unified description for the growing season correlation of the thermodynamic anomalies with opaque cloud and lagged precipitation, as shown in Table 3 and adapted from [16]. We retain the precipitation anomalies for four months.

**Table 3.** Standardized multiple regression coefficients for the May to August (MJJA) growing season merge of 2466 months. For coefficients: plain text represents $p < 0.01$ (>99%); italic represents $0.01 \leq p < 0.05$, and coefficients are omitted for $p > 0.05$.

| Variable | A ($\delta OPAQ_m$) | B ($\delta PR0$) | C ($\delta PR1$) | D ($\delta PR2$) | E ($\delta PR3$) | $R^2$ |
|---|---|---|---|---|---|---|
| $\delta T_x$ | $-0.95 \pm 0.02$ | $-0.07 \pm 0.02$ | $-0.16 \pm 0.02$ | | | 0.58 |
| $\delta T_m$ | $-0.67 \pm 0.02$ | $0.03 \pm 0.02$ | $-0.10 \pm 0.02$ | | | 0.43 |
| $\delta T_n$ | $-0.34 \pm 0.02$ | $0.18 \pm 0.02$ | | *$0.04 \pm 0.02$* | | 0.13 |
| $\delta DTR$ | $-0.61 \pm 0.01$ | $-0.26 \pm 0.01$ | $-0.15 \pm 0.01$ | $-0.05 \pm 0.01$ | $-0.03 \pm 0.01$ | 0.73 |
| $\delta RH_n$ | $0.59 \pm 0.01$ | $0.37 \pm 0.01$ | $0.23 \pm 0.01$ | $0.09 \pm 0.01$ | $0.03 \pm 0.01$ | 0.69 |
| $\delta RH_m$ | $0.53 \pm 0.01$ | $0.32 \pm 0.01$ | $0.24 \pm 0.01$ | $0.11 \pm 0.01$ | $0.04 \pm 0.01$ | 0.61 |
| $\delta RH_x$ | $0.38 \pm 0.02$ | $0.20 \pm 0.02$ | $0.20 \pm 0.01$ | $0.10 \pm 0.01$ | *$0.04 \pm 0.01$* | 0.36 |
| $\delta DRH$ | $-0.22 \pm 0.01$ | $-0.18 \pm 0.01$ | $-0.03 \pm 0.01$ | | | 0.26 |
| $\delta P_{LCLx}$ | $-0.76 \pm 0.02$ | $-0.42 \pm 0.02$ | $-0.31 \pm 0.01$ | $-0.13 \pm 0.01$ | $-0.05 \pm 0.01$ | 0.68 |
| $\delta P_{LCLm}$ | $-0.55 \pm 0.01$ | $-0.30 \pm 0.01$ | $-0.25 \pm 0.01$ | $-0.12 \pm 0.01$ | $-0.04 \pm 0.01$ | 0.62 |
| $\delta P_{LCLn}$ | $-0.30 \pm 0.01$ | $-0.15 \pm 0.01$ | $-0.16 \pm 0.01$ | $-0.08 \pm 0.01$ | $-0.03 \pm 0.01$ | 0.36 |
| $\delta DP_{LCL}$ | $-0.46 \pm 0.01$ | $-0.27 \pm 0.01$ | $-0.15 \pm 0.01$ | $-0.05 \pm 0.01$ | | 0.58 |
| $\delta\theta_{Ex}$ | $-0.55 \pm 0.02$ | $0.28 \pm 0.02$ | $0.08 \pm 0.02$ | $0.12 \pm 0.02$ | | 0.21 |
| $\delta\theta_{Em}$ | $-0.42 \pm 0.02$ | $0.30 \pm 0.02$ | $0.09 \pm 0.02$ | $0.11 \pm 0.02$ | | 0.17 |
| $\delta\theta_{En}$ | $-0.22 \pm 0.02$ | $0.34 \pm 0.02$ | $0.09 \pm 0.02$ | $0.11 \pm 0.02$ | | 0.13 |
| $\delta D\theta_E$ | $-0.32 \pm 0.01$ | $-0.06 \pm 0.01$ | | | | 0.37 |
| $\delta Q_m$ | $-0.06 \pm 0.02$ | $0.41 \pm 0.02$ | $0.22 \pm 0.02$ | $0.16 \pm 0.02$ | | 0.22 |

Table 3 shows that only some anomalies, such as $\delta DTR$, $\delta RH_n$, $\delta RH_m$, $\delta P_{LCLx}$ with high $R^2$ values, are correlated with precipitation anomalies going back three months. As in Table 2, the OPAQ coefficients A are typically the largest, except notably for $\delta Q_m$.

The first groups are the regression coefficients for the temperature anomalies, $\delta T_x$, $\delta T_m$, $\delta T_n$, and $\delta DTR$, which were all standardized by the SD of $\delta T_m$. The fit represented by $R^2$ is largest for DTR, and it decreases from $\delta T_x$ to $\delta T_n$. All the temperature variable anomalies show a strong inverse correlation with opaque cloud anomalies that reflect the downward SW radiation. The warm season is dominated by negative SWCF as shown in Figure 3. The negative values of A decrease from $\delta T_x$ to $\delta T_n$. $\delta DTR$ has a negative correlation to both cloud anomalies, and to the precipitation anomalies going back three months. Note that because all the temperatures were standardized by the SD of $\delta T_m$, the coefficients for the diurnal range are the difference of the corresponding coefficients for the maximum and minimum. For example, $A(\delta DTR) = -0.61 = A(\delta T_x) - A(\delta T_n)$, and $B(\delta DTR) = -0.26 = B(\delta T_x) - B(\delta T_n)$ (rounded to two significant figures). We see that the coefficients B change sign in the sequence from $\delta T_x$ to $\delta T_m$ to $\delta T_n$. We also see that $T_m$ falls strongly with cloud, but its coupling to precipitation is weak because the coefficients B and C have opposite signs. This regression analysis clearly shows that mean temperature anomalies, $\delta T_m$, are strongly coupled to cloud, and therefore solar forcing, but rather weakly to precipitation, while $\delta DTR$ (and $\delta T_x$) decrease with both cloud and precipitation. We cannot infer causality from multiple regressions, but negative B for $\delta T_x$ is consistent with evaporation from moist soils reducing $T_x$, and the positive B for $\delta T_n$ is consistent with the fact that under wetter conditions, the fall of $T_n$ at night is limited by saturation.

The next group are the four RH anomalies, $\delta RH_x$, $\delta RH_m$, $\delta RH_n$, and $\delta DRH$. For the first three, the regression coefficients show that positive RH anomalies are correlated with positive cloud and precipitation anomalies, and the coefficients are significant for both the present and three past months. The coefficients for $\delta DRH$ are negative because $\delta RH_n$ increases faster with cloud and precipitation than $\delta RH_x$, and the coefficients are significant for only one past month. The $R^2$ fit decreases monotonically from the afternoon minimum $\delta RH_n$ to $\delta RH_m$ to the sunrise maximum $\delta RH_x$ to $\delta DRH$. The diurnal cycle of T and RH have an inverse dependence on opaque cloud, reaching $T_x$ and $RH_n$ in the afternoon at the same time [16]. This is related to the fact that mixing ratio Q is tightly constrained by BL transports, which we will discuss in Section 4.3. However, over land, near-surface RH is constrained by the availability of soil moisture for evaporation from bare soil and transpiration (which is often modeled as a stomatal resistance to evaporation [22,23]. Soil moisture anomalies are related in turn to precipitation anomalies. We see that afternoon $RH_n$ and mean $RH_m$ anomalies have a strong positive correlation to precipitation anomalies, and a large $R^2$. However, $RH_x$, which increases with precipitation, is limited if the surface saturation is reached and dew forms before sunrise. Because the latent heat release slows the temperature fall, it is consistent that $RH_x$ and $T_n$ anomalies are both positively coupled to wetter precipitation anomalies for the current month (coefficient B).

The third group in Table 3 is the four $P_{LCL}$ anomalies: $P_{LCLx}$ is generally representative of afternoon cloud-base [9]. $P_{LCL}$ has a strong dependence on RH and a weak dependence on T, and we see that negative $P_{LCL}$ anomalies are coupled to positive cloud and precipitation anomalies. The coefficients are largest for afternoon $\delta P_{LCLx}$, for which $R^2$ is high. The coefficients for $\delta P_{LCLx}$, $\delta P_{LCLm}$, and $\delta P_{LCLn}$ are all 99% significant for both the present and three past months, showing that cloud-based anomalies have a long memory of precipitation anomalies in the growing season.

The fourth group in Table 3 shows the coefficients for $\delta\theta_{Ex}$, $\delta\theta_{Em}$, $\delta\theta_{En}$, and $\delta D\theta_E$. The first three show the decrease with increased cloud, but an increase with precipitation. The $R^2$ values are small, even though the coefficients have 99% confidence. The diurnal range of $\theta_E$ is dominated by the dependence of DTR on opaque cloud. The two afternoon anomalies, $\delta\theta_{Ex}$ and $\delta P_{LCLx}$, are related to moist convective instability, which is favored by a higher $\theta_{Ex}$ and a lower cloud base.

The diurnal variation of the mixing ratio, Q, has a double maxima and minima, which we will show in Section 4.3, and so Table 3 shows only the coefficients for the mean anomaly $\delta Q_m$. The $R^2$ fit is much smaller for Q than for RH. The negative correlation to opaque cloud is small, because T and RH have an inverse diurnal dependence on cloud. The positive correlation to precipitation anomalies goes back two months, consistent with positive precipitation anomalies increasing evapotranspiration.

Table 3 summarizes the multiple regression correlation coefficients between warm season near-surface variables and opaque cloud and lagged precipitation, and gives a quantitatively useful target for the evaluation of the coupled processes in models. Two important conceptual results emerge for the monthly mean climate on the Canadian Prairies. Afternoon anomalies of $\delta T_x$, $\delta RH_n$, $\delta P_{LCLx}$ are strongly correlated to opaque cloud anomalies. Correlation with precipitation anomalies are weaker, but stretch back for three past months for these key variables. Anomalies of $Q_m$ are coupled to precipitation anomalies with memory of two months past, but they have weak correlations to opaque cloud.

### 4.3. Growing Season Coupling of the Diurnal Cycle to Precipitation and Cloud

Figure 6 showed the very tight coupling in the warm season between opaque cloud and the diurnal range of key thermodynamic variables on daily timescales. Table 3 used multiple linear regression to show the correlation of the monthly anomalies of thermodynamic variables to anomalies of opaque cloud and precipitation. Table 3 confirms the strong correlation with opaque cloud, but shows that the coefficients for the lagged precipitation anomalies differ considerably for different variables.

For a graphical representation [16] we approximate by defining a weighted precipitation anomaly $\delta PRwt$, based on precipitation for just the current month and the past month:

$$\delta PRwt = 0.6 * \delta PR0 + 0.4 * \delta PR1 \tag{17}$$

This simplification, with this choice of coefficients in the ratio of 1.5, captures much of the precipitation dependence for the variables that have the highest $R^2$, such as DTR, $RH_n$, and $P_{LCLx}$, because these have the ratio of the coefficients $B/C \approx 1.5$ in Table 3.

The x-axis of Figure 7 is 0.1 bins of $OPAQ_m = \delta OPAQ_m + 0.46$, where 0.46 is the mean opaque cloud over all the months. For each MJJA month (total 2466 months), we computed the weighted anomaly $\delta PRwt$ from (17), and added the MJJA mean precipitation rate of 1.8 mm d$^{-1}$ to give $PRwt = \delta PRwt + 1.8$. We then stratified the data into three ranges of PRwt of <1.2 mm d$^{-1}$; 1.2 to 2 mm d$^{-1}$, and >2 mm d$^{-1}$, which have mean values of 0.9, 1.6, and 2.6 mm d$^{-1}$. There are (531, 1103, 832) months in these three PRwt bins. To generate Figure 7, we compute for each variable bin, the mean and standard error (SE) of the anomalies, and add back the MJJA variable means.

Figure 7 (top-left) shows DTR and its components, $T_x$ and $T_n$, the top-right shows DRH, $RH_x$ and $RH_n$, the bottom-left shows $D\theta_E$, $\theta_{Ex}$, and $\theta_{En}$ and the bottom-right is $DP_{LCL}$, $P_{LCLx}$, and $P_{LCLn}$. The strong dependence on opaque cloud, seen in Figure 6, clearly dominates most of these climate variables, since T falls and RH increases with increasing cloud. This is turn is connected to the weak dependence of Q on cloud (Table 3). The color scheme is red and blue, respectively, for the dry and wet precipitation bins. As PRwt falls, DTR increases faster than $T_x$.

Figure 7 (top-right) shows that $RH_x$ and $RH_n$ (and $RH_m$, not shown) increase with both cloud and PRwt, but because afternoon $RH_n$ increases faster than $RH_x$, DRH decreases with increasing PRwt. Note the rise of $RH_x$ with PRwt towards saturation. If $RH_x$ reaches saturation at the surface on individual days, condensation of dew and the release of latent heat limit the fall of $T_n$.

Figure 7 (bottom panels) show the variables that determine the BL coupling to clouds and precipitation. Afternoon $P_{LCLx}$ and $\theta_{Ex}$ determine the cloud-base height and moist adiabat. Both $\theta_{Ex}$ and $\theta_{En}$ increase with PRwt, but the diurnal range $D\theta_E$ depends primarily on cloud instead of precipitation, as shown in Table 3. All of the $P_{LCL}$ variables decrease with increasing PRwt. The sunrise minimum of $P_{LCLn}$ falls with PRwt, as the surface moves towards saturation. Thus, higher precipitation, which we can loosely associate with increased daytime evapotranspiration (ET), corresponds with a lower monthly mean cloud base and a higher $\theta_E$ in the afternoon, which would both favor increased convective instability.

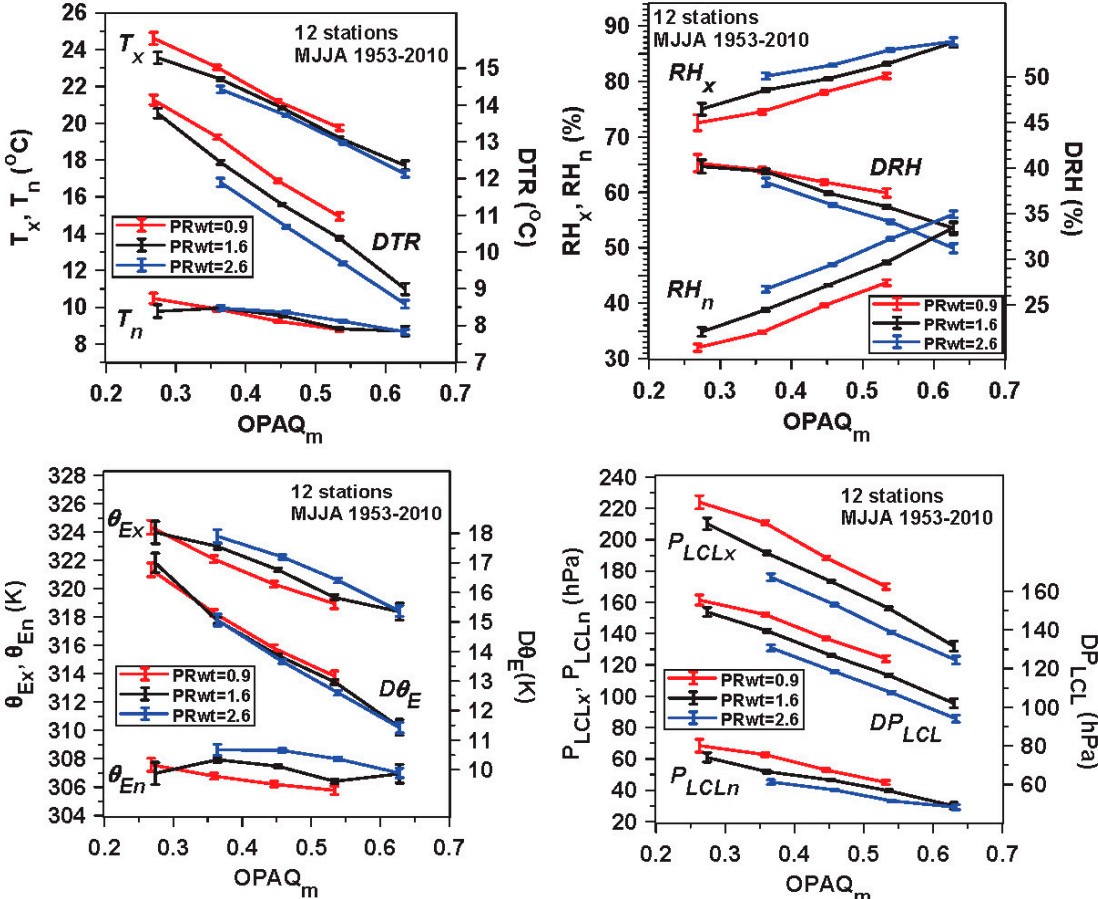

**Figure 7.** Coupling between DTR, $T_x$ and $T_n$ (**top-left**), (**top-right**) difference in relative humidity (DRH), $RH_x$ and $RH_n$, (**bottom-left**) $D\theta_E$, $\theta_{Ex}$ and $\theta_{En}$ and (**bottom-right**) $DP_{LCL}$, $P_{LCLx}$ and $P_{LCLn}$ and opaque cloud fraction and weighted precipitation in mm d$^{-1}$ (adapted from [16]).

In the warm season on the Prairies, the diurnal cycle of mixing ratio Q has two maxima and minima, except under cloudy conditions [9,14,16]. We can graph this dependence on anomalies of opaque cloud cover, $\delta OPAQ_m$, and weighted precipitation anomalies, $\delta PRwt$, from Equation (17) in mm d$^{-1}$, from the MJJA growing season merge of 2466 months.

Figure 8 (left panel) shows the stratification by $\delta OPAQ_m$ into four ranges: $\delta OPAQ_m < -0.08$; $-0.08$ to 0; 0 to 0.08, and >0.08, based on the SD of $\delta OPAQ_m \approx 0.08$. There are (371, 839, 909, 347) months in these respective bins. We averaged in bins the diurnal cycle of the anomalies from the station monthly means, calculate the SE, and added back the 12-station MJJA mean of Q. The legend shows the mean value for each $\delta OPAQ_m$ bin, and in parentheses the corresponding mean of $\delta PRwt$. As the mean $\delta OPAQ_m$ increases from $-0.12$ to $+0.12$, the mean $\delta PRwt$ increases from $-0.43$ to $+0.42$ mm d$^{-1}$. We have binned by $\delta OPAQ_m$, but mean OPAQm and precipitation increase together (Figure 6). The small increase in $Q_m$ with $\delta OPAQ_m$ is consistent with Table 3.

The sunrise minimum of Q occurs at the minimum temperature, when the night-time BL is shallow with a strong temperature inversion. As the surface net radiation turns positive after sunrise, it drives increasing surface sensible and latent heat fluxes. This warms and moistens a shallow ML trapped beneath the stable nocturnal inversion, and there is a steep rise of Q. When the surface potential temperature reaches that of the top of the capping inversion in mid-morning, the ML deepens more rapidly, typically mixing into a deep drier residual ML from the previous day, so that Q falls to the afternoon minimum. With less cloud and more solar forcing, the ML can grow deeper, and mix with more dry air from above, so both the morning rise and mid-day fall of Q are larger in Figure 8.

Finally Q rises again to an evening maximum as the surface layer cools and decouples from the deep BL, while ET continues.

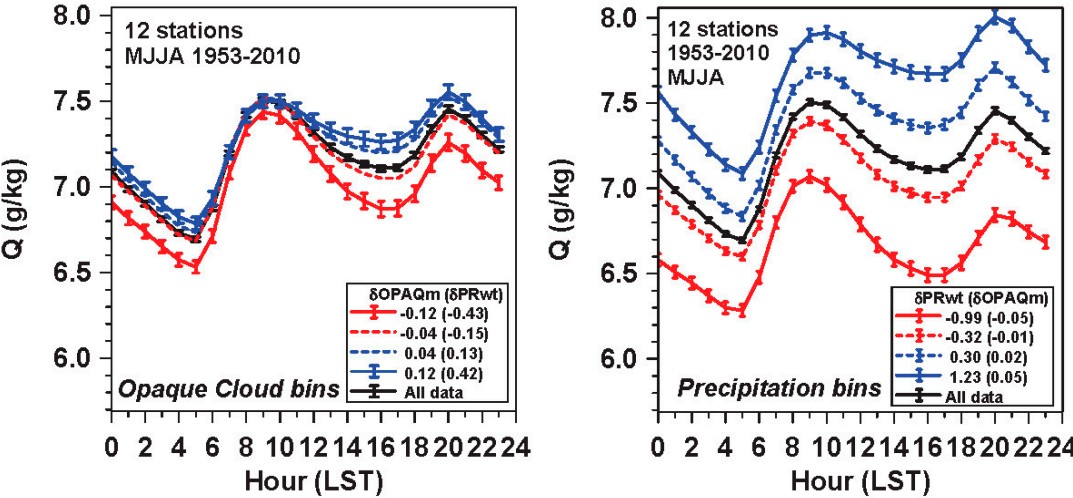

**Figure 8.** Dependence of diurnal cycle of the mixing ratio (Q) on opaque cloud bins (**left**) and weighted precipitation bins (**right**) [16].

Figure 8 (right panel) is the corresponding partition into four ranges of weighted precipitation anomalies: $\delta PRwt < -0.7$; $-0.7$ to $0$; $0$ to $0.7$, and $>0.7$ mm d$^{-1}$, based on the SD of $\delta PRwt \approx 0.7$ mm d$^{-1}$. There are (387, 961, 745, 373) months in these respective bins. The legend shows the mean value for each $\delta PRwt$ bin, and again in parentheses the corresponding mean of $\delta OPAQ_m$. With increasing $\delta PRwt$, there is a large upward shift of the mean diurnal cycle of Q, as $Q_m$ increases with precipitation anomalies, which we can associate with increased soil moisture and ET. As the mean $\delta PRwt$ increases from $-0.99$ to $+1.23$ mm d$^{-1}$, mean $\delta OPAQ_m$ increases from $-0.05$ to $+0.05$, and the fall of Q from mid-morning maximum to afternoon minimum is reduced as in the left panel.

Clearly, we are dealing with a fully coupled system, but Figure 3 shows that climatologically, while the amplitude of the diurnal cycle of Q increases a little with reduced cloud cover (presumably increased solar forcing and vertical mixing), there is a large upward shift in the diurnal cycle with increased weighted precipitation, presumably from increased ET.

## 5. Seasonal Climate Issues

### 5.1. Seasonal Extraction of Surface Total Water Storage

We made a simplified estimate of the growing season water and energy budgets of the Prairies in [12], using fits between opaque cloud and cloud forcing (see Section 3.3) to estimate the surface radiation budget. We calculated the seasonal change in total water storage from the GRACE, using the gridded $1 \times 1$ degree monthly land mass grids [7,8], version JPL-RL05.DSTvSCS1401, for liquid water equivalent thicknesses for the decade of 2002–2012. This allowed us to link the draw-down of total water storage (TWS) [24,25] to precipitation anomalies during the growing season. We will present a brief overview, referring to [12] for discussion of some of the uncertainties.

Figure 9 shows the mean annual cycle of the monthly anomalies of TWS from the annual mean, with the small 2002–2012 trend removed. Although the data have been interpolated to a $1 \times 1$ degree grid for user convenience, the effective spatial resolution of the GRACE data is about 300 km, so we averaged the data over the Prairie regions of the three provinces. The amplitude of the mean annual cycle is 90 mm. We computed ΔTWS:MJJA, the growing season drawdown of TWS for MJJA, as the difference between 31 August (mean of August and September) and 1 May (mean of April and May) for each year and province. Mean drawdown is ΔTWS:MJJA = $-79$ mm.

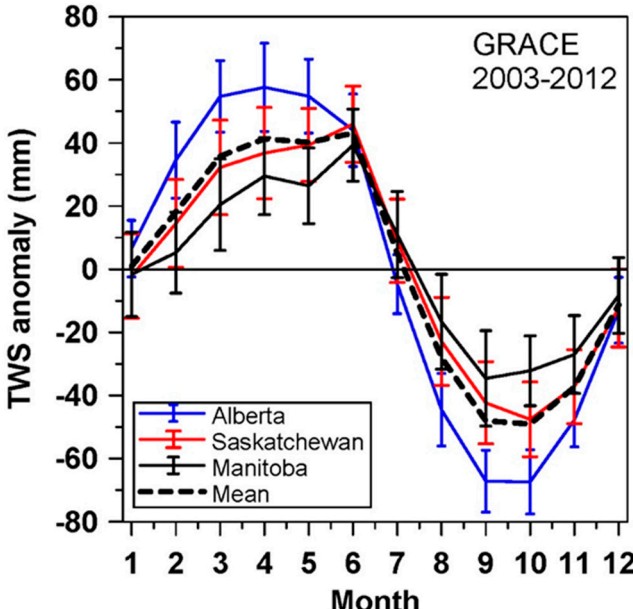

**Figure 9.** The mean annual cycle of Gravity Recovery and Climate Experiment (GRACE) total water storage (TWS) anomaly by province (from [12]).

We regressed the anomalies of ΔTWS:MJJA for the years 2002–2012 onto the corresponding anomalies of MJJA precipitation, derived from the monthly archive (see [26]: http://ec.gc.ca/dccha-ahccd/) of the second generation adjusted precipitation dataset. The linear regression fit [12] with $R^2$ = 0.56, is:

$$\Delta\text{TWS:MJJA} = -0.59(\pm 0.08) + 0.56(\pm 0.09)\ \delta\text{Precip(MJJA)} \tag{18}$$

The mean value of Precip(MJJA) for 2002–2012 (details in [12]) is 2.32 mm d$^{-1}$, while the mean drawdown of TWS is 0.59 mm d$^{-1}$ for δPrecip(MJJA) = 0; or 25% of the mean precipitation. Thus Equation (18) shows that, as δPrecip(MJJA) decreases from +1 to −1 mm d$^{-1}$, ΔTWS:MJJA increases from near zero to −1.15 mm d$^{-1}$, which corresponds to −141 mm over the 123 day growing season. The coupling coefficient of 0.56($\pm$0.09) in Equation (18) is effectively a 56 $\pm$ 9% damping coefficient for precipitation anomalies in the growing season by changes in the drawdown of TWS. This drawdown of stored water in the growing season means that ET > precipitation, and this difference increases in dry summers.

*5.2. Impact of Land-Use Change on Growing Season Climate*

In recent decades, there has been a major change in land use across the Canadian Prairies, as more than five million hectares of summer fallow have been converted to continuous cropping [10]. This large increase in the area of cropland has increased summer transpiration, which in turn have reduced the maximum temperatures in the growing season over the Prairies [27]. Other analyses of US Midwest summer temperature maxima also show a cooling from land-use change to cropland [28] and cropland intensification [29].

Figure 10 summarizes the long-term climate impact of the reduction of summer fallow in Saskatchewan [10,15]. The left panel shows the land-use trends in total cropland, pasture, and summer fallow around five climate stations in Saskatchewan. We generated local averages of the ecodistrict crop data [10] within the 50 km radius circles around each station, as shown in Figure 1. The climate station time-series was split into two periods: a longer historic period, 1954–1991, when summer fallow cover was large (although slowly decreasing), and a recent 20-year period, 1992–2011, when summer fallow has fallen rapidly to its present low value [30].

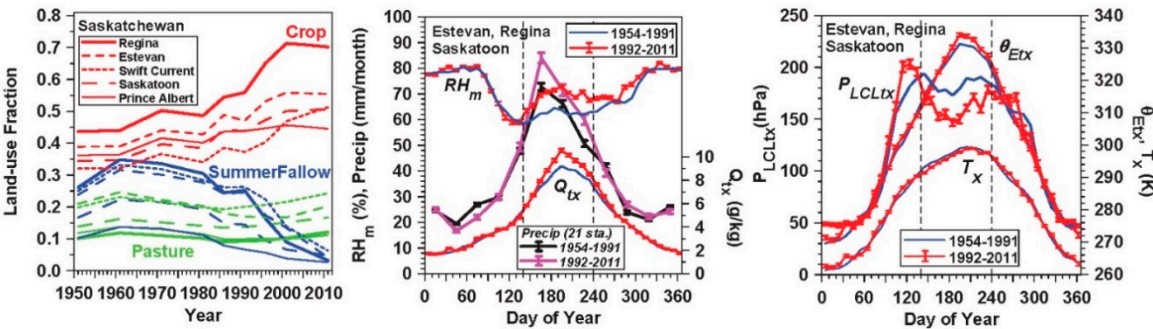

**Figure 10.** Long term trends in total cropland, pasture, and summer fallow around five climate stations in Saskatchewan (**left**); (**center**) $RH_m$, $Q_{tx}$ and mean precipitation in southern Saskatchewan, and (**right**) mean changes in annual cycle of $T_x$, $P_{LCLtx}$, and $\theta_{Etx}$ for Saskatoon, Regina, and Estevan (adapted from [10,15]).

The center panel shows the changes between the two time periods. For $RH_m$ and mixing ratio $Q_{tx}$ at the time of the afternoon $T_x$. We averaged the 10-day means from Saskatoon, Regina, and Estevan, the three southern stations in Saskatchewan with complete datasets [10]. We show the standard errors (SE) of the difference between the two mean time series as an indication of significance. For precipitation, which has much more variability than temperature and humidity, we used monthly precipitation for the 21 stations in Saskatchewan south of 53.22° N, from the second generation-adjusted precipitation dataset [26]. Error bars are the SE of each monthly mean. The right panel shows the corresponding changes in the annual cycle of $T_x$, $P_{LCLtx}$, and $\theta_{Etx}$ between the two time periods.

It is clear that there are significant changes in the growing season climate between the historic period, 1953–1991, and the more recent period since 1992. The vertical dashed lines mark the period $140 \leq DOY < 240$ (20 May–27 August) considered in [10] to be representative of the crop growing season. Over this time window, the growing season is cooler since 1992, with a drop of $T_x$ of $-0.93 \pm 0.09$ K, and significantly moister with a rise of ($RH_m$, $Q_{tx}$) of ($6.9 \pm 0.2\%$, $0.70 \pm 0.04$ gkg$^{-1}$). There is a corresponding fall of the $P_{LCLx}$ of $22.3 \pm 1.1$ hPa, and a small rise of $\theta_{Etx}$ of $1.1 \pm 0.2$ K, both at the time of afternoon $T_x$. There is also an increase of summer (June, July, and August) precipitation of $25.9 \pm 4.6$ mm.

It is reasonable to conclude that we are seeing the fully coupled response to the large shift from summer fallow to intensive cropping. It seems that more intensive agriculture has increased transpiration, which has cooled and moistened the growing season climate, lowering the cloud-base and increasing the equivalent potential temperature. The coupled increase of Q with precipitation is consistent with Figure 8; and the increase of summer precipitation is consistent with the increase in moist instability, represented by lower afternoon $P_{LCLx}$ and higher $\theta_{Etx}$. Not shown here is a distribution shift in cloud frequency: with 6% fewer days with 2–4 tenths, and 7% more with 7–10 tenths cloud cover [10].

### 5.3. Warm Season Atmospheric and Surface Energy Budgets

The Prairie climate data have no surface nor TOA fluxes, so we will use flux data from ERA-Interim to put our analyses in the context of the atmospheric and surface energy budgets for the MJJA warm season. We used gridpoint data co-located with the four climate stations Estevan, Regina, Saskatoon, and Prince Albert in Saskatchewan that were used in [31] to analyze the near-surface biases of temperature in ERA-Interim. We computed the terms in the atmospheric TOA and surface energy budget, first on a daily basis, and then for each MJJA station-year, dropping a few days in May with surface snow cover. Figure 11 is a simplified graphical representation of the key flux terms for this region of the central Prairies from 49.2 to 53.2° N.

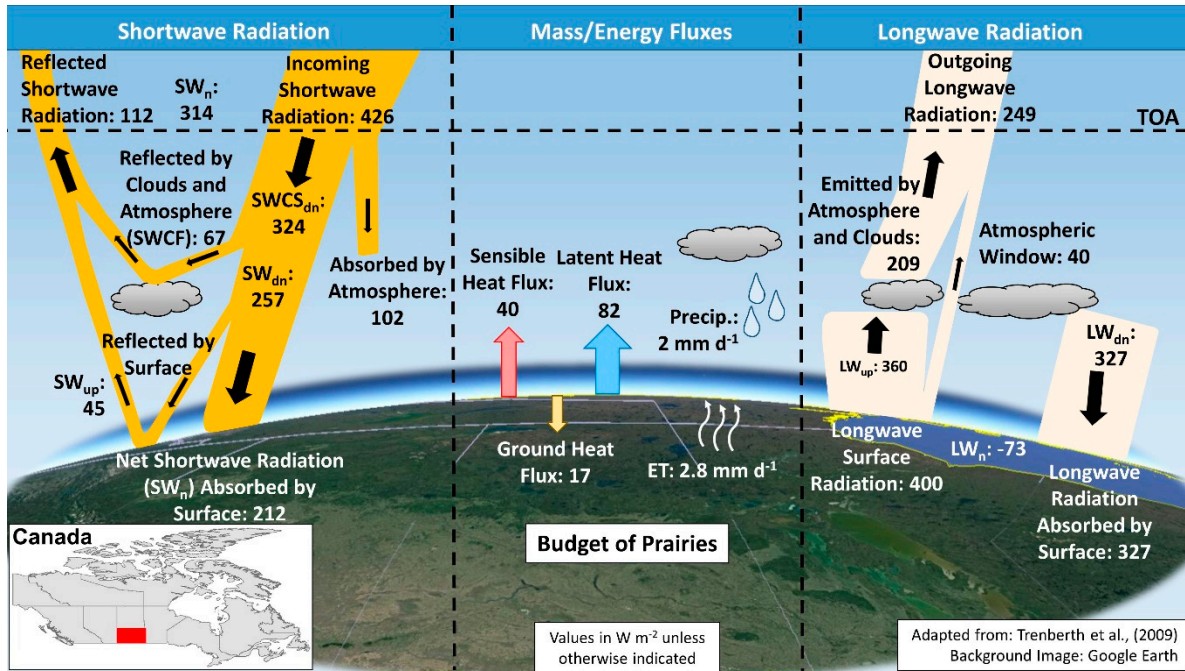

**Figure 11.** Schematic for surface and top-of-atmosphere (TOA) energy budgets for Prairie region of southern Saskatchewan.

On the left is the SW budget terms. Part of the incoming TOA solar flux is absorbed by the atmosphere, part reflected by clouds and atmospheric aerosols, and part is reflected by the surface; so that surface $SW_n$ = 212 W m$^{-2}$ is about half the TOA downward flux. On the right is the LW budget terms, where the upward surface LW emission of 400 W m$^{-2}$ is mostly absorbed by the atmospheric greenhouse gases, primarily water vapor and $CO_2$, as well as by clouds. About 10% escapes to space through the atmospheric infrared window. Clouds and atmosphere re-emit to space, and back to the surface, giving a surface $LW_n$ = −73 Wm$^{-2}$. The resulting surface $R_n$ = 139 W m$^{-2}$ balances the upward sensible and latent heat fluxes (40 and 82 W m$^{-2}$ respectively), and the growing season warming of the ground of 17 W m$^{-2}$, which is probably too large [31]. For the MJJA surface water budget, precipitation is 2 mm d$^{-1}$ and ET is 2.8 mm d$^{-1}$. ET exceeds precipitation in the growing season, because of the substantial drawdown of surface water storage, as discussed in Section 5.1. The TOA $SW_n$ > outgoing LW for the MJJA warm season.

Table 4 stratifies the 140 station years of data into ECA bins, showing how reflective cloud cover changes the TOA and surface energy budgets. The mean values across all the data in bold are those that are shown as rounded values in Figure 11. The model sign convention is downward fluxes are positive and upward fluxes are negative. The number of station-years in each bin, denoted by K, is not uniform, nor is the distribution across the four stations. Nonetheless, there is a uniform progression as reflective cloud (ECA) increases: TOA $SW_n$, surface $SW_{dn}$, net radiation, $R_n$, the sensible heat flux, SH, Bowen ratio, BR, and the 2 m temperatures all decrease, while the latent heat flux, LH, and precipitation increase. There is compensation between the surface $SW_n$ and $LW_n$ fluxes as reflective cloud changes, which reduces the change in $R_n$. If there is less reflection by clouds, $SW_{dn}$ increases, and this warms the surface, and $LW_{up}$ increases. Because the decrease of cloud cover and precipitation are coupled, and ET changes much less than precipitation, we see that the surface BR increases by a factor of two as cloud cover and precipitation decrease.

**Table 4.** Surface and TOA fluxes for the MJJA warm season from ERA-Interim for 1979–2013, stratified by ECA, derived from the gridpoints for Estevan, Regina, Saskatoon, and Prince Albert, SK. Mean data in bold are shown in Figure 11.

| ECA | K | TOA SW$_{dn}$ | TOA SW$_n$ | Surf SWCS$_{dn}$ | Surf SW$_{dn}$ | SWCF | Surf SW$_{up}$ | Surf SW$_n$ | Surf Albedo | TOA LW$_{up}$ | Surf LWup | Surf LW$_{dn}$ | Surf LW$_n$ |
|---|---|---|---|---|---|---|---|---|---|---|---|---|---|
| 0.146 | 11 | 427 | 326 | 327 | 279 | −48 | −54 | 225 | 0.19 | −257 | −412 | 328 | −84 |
| 0.178 | 34 | 427 | 320 | 325 | 268 | −58 | −49 | 218 | 0.18 | −252 | −406 | 329 | −77 |
| 0.203 | 47 | 426 | 315 | 324 | 258 | −66 | −45 | 213 | 0.17 | −250 | −400 | 327 | −74 |
| 0.230 | 32 | 424 | 310 | 322 | 248 | −74 | −40 | 208 | 0.16 | −247 | −396 | 326 | −69 |
| 0.278 | 16 | 427 | 299 | 323 | 233 | −90 | −38 | 195 | 0.16 | −242 | −389 | 325 | −64 |
| **0.207** | **140** | **426** | **314** | **324** | **257** | **−67** | **−45** | **212** | **0.17** | **−249** | **−400** | **327** | **−73** |

| ECA | K | R$_n$ | SH | LH | BR | G | PRECIP | ET | Runoff | T$_x$ | T$_m$ | T$_n$ | DTR |
|---|---|---|---|---|---|---|---|---|---|---|---|---|---|
| 0.146 | 11 | 141.6 | −53.2 | −71.8 | 0.78 | 16.7 | 1.13 | −2.48 | 0.07 | 24.4 | 18.5 | 12.1 | 12.3 |
| 0.178 | 34 | 141.4 | −46.1 | −77.4 | 0.62 | 17.9 | 1.69 | −2.67 | 0.04 | 23.0 | 17.4 | 11.2 | 11.8 |
| 0.203 | 47 | 139.6 | −41.1 | −81.5 | 0.53 | 17.0 | 1.91 | −2.82 | 0.06 | 22.0 | 16.5 | 10.5 | 11.5 |
| 0.230 | 32 | 138.3 | −33.3 | −88.9 | 0.39 | 16.1 | 2.18 | −3.07 | 0.13 | 21.0 | 15.8 | 10.1 | 11.0 |
| 0.278 | 16 | 130.8 | −30.9 | −84.5 | 0.37 | 15.4 | 2.92 | −2.92 | 0.13 | 19.4 | 14.5 | 9.2 | 10.3 |
| **0.207** | **140** | **138.9** | **−40.3** | **−81.8** | **0.52** | **16.8** | **1.97** | **−2.83** | 0.08 | 21.9 | 16.5 | 10.6 | 11.4 |

The reanalysis is a fully coupled model representation of the Prairie climate for this 35-year period, mapped here in terms of changing ECA. On average ET exceeds precipitation by 0.86 mm d$^{-1}$, which is largely consistent with the budget analysis in Section 5.1, based on the GRACE data. There is a substantial MJJA drawdown of about 80 mm of soil water (not shown), because the model has four soil layers to a total depth of 2.89 m with a dynamic range of water storage of 150 mm per meter between field capacity and permanent wilting point. However, the model soil water budget has additional increments, because the soil moisture reanalysis uses observed 2 m values of temperature and humidity and satellite estimates of soil moisture [32] to minimize the model 2 m temperature forecast errors. Despite this analysis correction, we found in [31] that on daily timescales, ERA-Interim in the warm season has a cold bias in $T_x$ and a warm bias in $T_n$, so that DTR is biased low; and these biases increase under clear skies. Table 4 stratifies four-month composites by ECA, so that it is not directly comparable to the daily analysis in [31], but it is likely that $T_x$ and $T_n$ in Table 4 have respectively cold and warm biases of about 1 °C.

## 6. Conclusions

We have reviewed progress in the quantitative understanding of the coupling between the land surface, clouds, precipitation, snow cover and the climate system that came from analysis of the long-term hourly Canadian Prairie dataset. This progress was possible because, along with conventional hourly measurements, trained observers have recorded opaque cloud fraction hourly across the Prairies for the past 60 years. These 24 daily estimates of opaque cloud are of sufficient quality such that they can be calibrated against BSRN data to obtain the climatology of the daily short-wave, long-wave, and total cloud forcing. This key radiative forcing of the surface energy budget has previously not been available for long-term climate datasets. We found that net cloud radiative forcing changes sign from negative in the warm season to positive in the cold season with snow cover. This transforms the coupling between cloud cover and the diurnal cycle between warm and cold seasons. In the warm season, maximum temperature increases with decreasing opaque cloud, while minimum temperature barely changes. In contrast in the cold season with snow cover, maximum temperature falls with decreasing cloud while minimum temperature falls even more steeply. Although our results are stratified by the observed opaque cloud cover, we show the regression Equations (10) and (11) that convert opaque cloud into cloud radiative forcing.

We have shown the many ways in which snow cover acts as a climate switch between two non-overlapping climate regimes, producing a systematic cooling of 10 °C or more. With fresh snow cover in November, the temperature falls 10 °C during the transition. Simply separating cold season days into those with and without snow cover shows that these are two distinct climates, again separated by 10 °C. The traditional merging of these two climates is unrepresentative, since typically the near-surface boundary layer also changes from unstable to stable with snow cover. As cloud cover falls, the climate cooling with snow increases. Mean cold season temperatures fall by almost 1.5 °C for each 10% increase in the fraction of days with snow cover.

In the warm season with no snow cover, the diurnal ranges of temperature, relative humidity, equivalent potential temperature, and the pressure height of the lifting condensation level all fall steeply with increasing opaque cloud cover. Remarkably, this tight coupling between diurnal range and cloud cover is almost unchanged from April to September, so we provide quadratic fits to the mean profiles. Given 600 station-years of hourly data, we are able to extract, perhaps for the first time, the relationship between cloud forcing and the warm season imbalance of the diurnal cycle. This imbalance over 24 h changes monotonically from a +2 °C warming and a $-6\%$ drying under clear skies, to a $-1.5$ °C cooling and 6% moistening under cloudy skies with precipitation. Correspondingly, there is a 24 h rise of $\theta_E$ and LCL under clear skies, and a fall of both under cloudy skies.

On daily timescales, the radiative forcing dominates, and the changing cloud forcing drives the changing diurnal cycle of the thermodynamic variables. On longer timescales, such as monthly and seasonal, precipitation anomalies, stored as soil moisture anomalies, change the ET and modify the

diurnal response. Since we know the cloud radiative forcing which is large on daily timescales, we can show statistically that the memory of water storage anomalies, from precipitation and the snowpack, goes back many months. The spring climatology for April shows the memory of snowfall back through the entire winter, and the memory in summer months goes back to the months of snowmelt.

In addition, we showed how the thermodynamic coupling of the diurnal cycle to the cloud forcing is modified by lagged precipitation anomalies. With reduced precipitation, the diurnal ranges of T, RH, and $P_{LCL}$, but not $\theta_E$, increase. With increased precipitation, afternoon $T_x$ falls a little, while $RH_n$ and $\theta_{Ex}$ increase and $P_{LCLx}$, representing cloud-base, falls.

Climatologically, the diurnal cycle of Q has a sunrise minimum, a rise to a mid-morning maximum while evaporation is trapped beneath the nocturnal inversion, then a fall to an afternoon minimum, as water vapor is rapidly transported upward into a deep daytime BL; followed by a second rise to an evening minimum as the surface cools and starts to uncouple from the mixed BL. We showed that while the amplitude of the diurnal cycle of Q increases a little with reduced cloud cover (presumably from increased solar forcing and vertical mixing), there is a large upward shift in the diurnal cycle of Q as weighted precipitation increases, presumably from increased evaporation.

The GRACE satellite data from a recent period show that the seasonal extraction of the surface total water storage is a large damping of the interannual variability of precipitation anomalies in the growing season. Over a range of precipitation anomalies of $\pm 1$ mm d$^{-1}$, the seasonal extraction of ground water increases from near-zero with high precipitation, to 1.15 mm d$^{-1}$ for low precipitation.

Also on seasonal timescales, the large land-use change on the Prairies from summer fallowing to intensive cropping, with the most rapid transition in the early 1990s, has led to a coupled climate response that has cooled and moistened the growing season, lowering cloud-base, increasing equivalent potential temperature, and increasing precipitation.

Finally, we compute the atmospheric energy and water budgets for four stations in Saskatchewan for the 35-y period of the ERA-Interim reanalysis, to show graphically the mean TOA and surface budgets that are representative of the Canadian Prairies for the MJJA growing season. We also show the dependence of these budgets on reflective cloud cover for this fully coupled model system.

More broadly, these long-term climate datasets from the Prairies can be used to improve the representation of clouds and land-surface processes in numerical forecast models. A preliminary step was the assessment of the biases in the surface diurnal cycle of temperature in ERA-Interim [31], but much more is possible. For example, knowledge of the biases is providing meteorological information for regions where no data were previously available. We plan to investigate the impact of these biases on model estimates of crop yields and greenhouse gas emission estimates using the DNDC model [33]. With the improved knowledge of the energy budget for the Prairies, we can now investigate which management practices could significantly change the energy budget and allow farmers opportunities to minimize their impact on climate change.

**Author Contributions:** Conceptualization, writing—original draft preparation, A.K.B.; writing—review and editing, A.K.B. and R.L.D.

**Funding:** This research was partially supported by Agriculture and Agri-Food Canada, contract number 3000662193, by the National Science Foundation under VT EPSCoR Grant No. NSF OIA 1556770, and with technical support from ECMWF.

**Acknowledgments:** We are very grateful to the Canadian station observers who have made hourly climate observations for many decades, and to the many co-authors who contributed to this series of papers.

**Conflicts of Interest:** The authors declare no conflict of interest.

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
