# Peer review of "Understanding Land–Atmosphere–Climate Coupling from the Canadian Prairie Dataset"

_environments, doi:10.3390/environments5120129_

Round 1
Reviewer 1 Report
p { margin-bottom: 0.25cm; ; }This review paper aims to reflect previous work by the authors regarding land surface – atmosphere coupling by using opaque cloud information and climatological data from 15 climate stations in the Canadian Prairie. Based on a 17-year radiation data set at one station climate coupling in terms of cloud radiative forcing on a diurnal and seasonal cycle has been conducted. Additionally, clear-sky fluxes obtained from ERA Interim reanalysis data were used to define the effective cloud albedo.
Comments:
1) Overall the manuscript is well written and, as shown by their previous work the authors have great experiences in this field. However, land surface – atmosphere coupling, cloud albedo effects, radiation budgets etc. are large fields and this should be more reflected and discussed with other studies in a much broader way.
2) It is hard to follow, which time periods were used for the analyses. Make it more transparent.
3) It is confusing whether the ensemble of the stations is used or individual stations. In the latter it is not clear why the respective station is analyzed. Be more concise.
Minor comments:
Figure 1: Despite the scale, the map should be provided with coordinates
Figure 2: The y-axes must have the same scale in order to be comparable.
Figure 4-10: The legends are very small and in parts not readable. Figures 4 and 10 need to be enlarged, so that all details can be seen.
Author Response
Reviewer Comments: Response Highlighted
1) Overall the manuscript is well written and, as shown by their previous work the authors have great experiences in this field. However, land surface – atmosphere coupling, cloud albedo effects, radiation budgets etc. are large fields and this should be more reflected and discussed with other studies in a much broader way.
We considered this issue before writing the paper and chose not to attempt a broad review of such a vast field. We have added text to make this clear.
“This paper is not a conventional review of the literature on land-atmosphere-climate coupling. Instead, it is a synthesis of our key conclusions from a series of Prairie papers [9-16].”
2) It is hard to follow, which time periods were used for the analyses. Make it more transparent.
This is clearly an issue, and the challenge of this paper because we have synthesized results from seven manuscripts addressing different issues.
We have opted to go through the subsections and improve the clarity of the text as you suggest. Specifically we clearly identify the data used, explain the many timescales and issues involved, and so the context of each subsection is clear.
3) It is confusing whether the ensemble of the stations is used or individual stations. In the latter it is not clear why the respective station is analyzed. Be more concise.
Understood. Text added with Table 1. We checked that data in each figure is identified, and other station examples in original papers are referenced, where a representative one was selected.
Minor comments:
Figure 1: Despite the scale, the map should be provided with coordinates
Lat-long markers added
Figure 2: The y-axes must have the same scale in order to be comparable.
You are right. We have redrafted Figure 2 (published in [14]), using just two ranges on the y-axis : one for all the nine months JFMJJASOD (21K), and a larger one (32K) for the 3 transition months, MAN, where there is a 10K separation with and without snow. Commented in text.
Figure 4-10: The legends are very small and in parts not readable. Figures 4 and 10 need to be enlarged, so that all details can be seen.
Fig 4 redrafted to remove fine print; increase font size
Fig 10 redrafted to increase font size
We also edited
Fig 7 font size increased
Fig 8 font size increased
Reviewer 2 Report
The authors analyse 60 years hourly Canadian Prairie data. They calibrate opaque cloud data against key radiative forcing variables. They show how snow cover acts as a climate switch between non-overlapping climate regimes. It is found that net cloud radiative forcing reverses sign from negative in the warm season to positive in the cold season. Maximum temperature increases with decreasing opaque cloud, while minimum temperature does not change much in the warm season. Maximum temperature decreases with decreasing cloud and minimum temperature decreases even more in the cold season with snow. On daily timescales the radiative forcing dominates, and the changing cloud forcing drives the changing diurnal cycle of the thermodynamic variables. On longer monthly and seasonal timescales, soil moisture anomalies change evapotranspiration and modify the diurnal response. The long-term climate datasets from the Prairies can be used to improve the representation of clouds and land-surface processes in numerical forecast models.
The manuscript is of major importance to the science community, and I recommend publication after minor revision. The manuscript needs to be structured in a more concise way starting with the abstract followed by the results‘ section 3. As the manuscript is written it is not easy to follow the different forcings on daily, monthly, and seasonal timescales. Further the many equations of the resultss‘ section need to be sorted and moved to the methods‘ section as well.
Lines 203 – 221: Please move these lines with the equations to methods‘ section.
Lines 233 – 246: Please move these line with the equations to the methods‘ section.
Lines 262 – 265: Please move these line with the equations to the methods‘ section, and explain why a quadratic fit is preferred.
There are many more of such paragraphs. Please also move these to the methods‘ section with a general paragraph about the regression analysis. The results of the slopes and coefficients of determination can be placed in the results‘ section in tables.
Author Response
Reviewer (Response highlighted)
The manuscript is of major importance to the science community, and I recommend publication after minor revision. The manuscript needs to be structured in a more concise way starting with the abstract followed by the results‘ section 3. As the manuscript is written it is not easy to follow the different forcings on daily, monthly, and seasonal timescales. Further the many equations of the resultss‘ section need to be sorted and moved to the methods‘ section as well.
Lines 203 – 221: Please move these lines with the equations to methods‘ section. DONE
Lines 233 – 246: Please move these line with the equations to the methods‘ section. DONE for 233-240
Lines 262 – 265: Please move these line with the equations to the methods‘ section, and explain why a quadratic fit is preferred.
We have removed the quadratic fits from the LW panels, but left the regression fits here for the discussion of Figure 4 – see below
There are many more of such paragraphs. Please also move these to the methods‘ section with a general paragraph about the regression analysis. The results of the slopes and coefficients of determination can be placed in the results‘ section in tables.
We examined these suggestions carefully.
We agree that the equations (3) to (9) should be in section 2 and we have moved them – thanks – this is clearer.
However equations (10) to (14) all describe fits to specific data in Figures 4, 5, 6. They are needed to explain these Figures, and they would not make sense to the reader in section 2.
We explored moving the regression analysis discussion at the beginning of section 4 to section 2 but this is also out of context. We concluded that the reader needed the complex details of this normalized multiple regression in section 4.
Instead we have opted to go through the subsections and improve the clarity of the text (as suggested by the other reviewer). Specifically explaining the many timescales and issues involved, and clearly identifying the data used, so the context of each subsection is clear. The challenge of this paper is that we have synthesized results from seven manuscripts addressing different issues.
We have improved/redrafted 6 figures at the request of the other reviewer